# IS SYNTHETIC IMAGE USEFUL FOR TRANSFER LEARNING? AN INVESTIGATION INTO DATA GENERATION, VOLUME, AND UTILIZATION

## ABSTRACT

Synthetic image data generation represents a promising avenue for training deep learning models, particularly in the realm of transfer learning, where obtaining real images within a specific domain can be prohibitively expensive due to privacy and intellectual property considerations. This work delves into the generation and utilization of synthetic images derived from text-to-image generative models in facilitating transfer learning paradigms. Despite the high visual fidelity of the generated images, we observe that their naive incorporation into existing real-image datasets does not consistently enhance model performance due to the inherent distribution gap between synthetic and real images. To address this issue, we introduce a novel two-stage framework called bridged transfer, which initially employs synthetic images for fine-tuning a pre-trained model to improve its transferability and subsequently uses real data for rapid adaptation. Alongside, We propose dataset style inversion strategy to improve the stylistic alignment between synthetic and real images. Our proposed methods are evaluated across 10 different datasets and 5 distinct models, demonstrating consistent improvements, with up to 30% accuracy increase on classification tasks. Intriguingly, we note that the enhancements were not yet saturated, indicating that the benefits may further increase with an expanded volume of synthetic data.

## 1 INTRODUCTION

Pre-training a model on a large-scale dataset and subsequently transferring it to downstream tasks has proven to be both a practical and effective approach to achieving exceptional performance across a variety of tasks (Sharif Razavian et al., 2014; Donahue et al., 2014). In the transfer learning pipeline, a model is initially trained on a source dataset and later fine-tuned on various downstream datasets (aka target datasets). Source datasets are typically large-scale, general-purpose, and publicly available, such as ImageNet-1K/21K (Deng et al., 2009). Conversely, downstream datasets are typically smaller and domain-specific.

Given the popularity of this paradigm, numerous studies have investigated the different factors that influence the performance of downstream tasks. For example, Azizpour et al. (2015) found that augmenting the network's depth resulted in better transfer learning performance than increasing its width. Similarly, Kornblith et al. (2019) identified a strong correlation between ImageNet accuracy and transfer learning performance. Other research, such as Dosovitskiy et al. (2020); Chen et al. (2020); Jia et al. (2022), has enhanced transfer learning by introducing novel network architectures or learning paradigms. Despite these advancements, little work has been done to investigate the potential benefits of leveraging additional data to facilitate the transfer learning process. This may be due to the fact that collecting real-world data for domain-specific downstream tasks is both expensive and fraught with privacy and usage rights concerns.

This work explores a novel approach to improving the transferability of ImageNet pre-trained models: data synthesis. Recent advancements in Generative AI have facilitated the synthesis of high-quality, photo-realistic images. Text-to-image models (Nichol et al., 2021; Ramesh et al., 2022; Saharia et al., 2022b), such as Stable Diffusion (StabilityAI, 2023), can generate millions of new

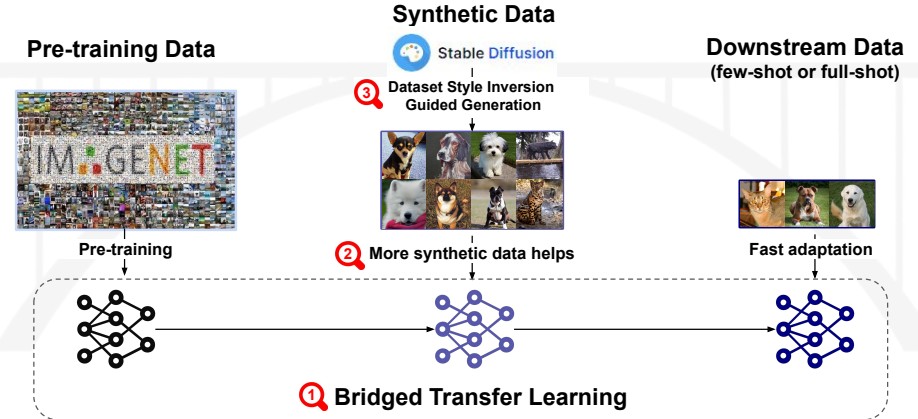

Figure 1: We present our bridged transfer learning framework, which initially employs synthetic images for fine-tuning a pre-trained model to improve its transferability and subsequently use real images for rapid adaption (Sec. 4.1). Under this framework, we also explore the influence of synthetic images volume (Sec. 4.2) and style alignment between real and style alignment between real and synthetic images, achieved through our proposed dataset style inversion technique (Sec.4.3).

images from text-based instructions. We leverage these text-to-image models to generate images from the target domain, thereby expanding the target dataset in a manner akin to data augmentation.

To comprehensively understand the impact of synthetic data generated by text-to-image models on ImageNet transfer learning tasks, we scrutinize three key factors through extensive experiments across 10 distinct downstream datasets, as also demonstrated in Figure 1:

1. *Data utilization.* We conduct a comparative analysis between the exclusive use of downstream real data, and the mixing of synthetic data into the real dataset, aiming to create an augmented dataset. Contrary to expectations, the mixing of real and synthetic data fails to improve transfer learning, often resulting in degraded accuracy. This implies that synthetic images may obscure the distribution of the real data. To effectively leverage synthetic images while circumventing this pitfall, we introduce a two-stage framework, termed Bridge Transfer. Initially, synthetic images are deployed to fine-tune the ImageNet pre-trained model, thereby optimizing its applicability for the downstream data. In the subsequent stage, real data is harnessed for fast adaptation from this fine-tuned model. Empirical findings suggest that Bridge Transfer fosters a more transferable model, resulting in consistent enhancements to the final performance of transfer learning.
2. *Data volume.* We test the volume of synthetic data from 500 per category to 3000 images per class. Our findings indicate a positive correlation between this increase and enhanced task performance. While the rate of performance improvement tapers off incrementally with the escalating volume of synthetic data, a saturation effect remains elusive.
3. *Data generation control.* The guidance scale serves as a crucial hyper-parameter in the Stable Diffusion model, orchestrating the alignment between the textual input and the resulting images. Our experiments with various guidance scales evidence their robustness within our Bridge Transfer framework. Moreover, we introduce Dataset Style Inversion (DSI), designed to further incentivize synthetic images to adhere to the style characteristic of the downstream dataset. DSI effectively concentrates the downstream dataset into a style token, subsequently employed to steer the image generation process. Empirical results reveal that DSI effectively fosters alignment between synthetic and real images, thereby enhancing the accuracy of transfer learning. This approach offers a notable computational efficiency advantage over traditional sample-wise textual inversion methods.

## 2 RELATED WORK

**Transfer Learning.** Transfer learning in image recognition starts with CNNs (Donahue et al., 2014; Chatfield et al., 2014; Sharif Razavian et al., 2014; Azizpour et al., 2015). It can be adapted to other domains like medical imaging (Mormont et al., 2018), language modeling (Conneau & Kiela, 2018), and various object detection and segmentation-related tasks (Ren et al., 2015; Dai et al., 2016; Huang et al., 2017). As discussed throughout this paper, several prior works (Kornblith et al., 2019; Zamir et al., 2018; Kolesnikov et al., 2020; Mahajan et al., 2018) have investigated factors

improving or otherwise affecting transfer learning performance. (Huang et al., 2019; Kolesnikov et al., 2020) have demonstrated scaling up the pre-training datasets and models can effectively lead to better transfer performance. Aside from CNNs, other architectures like ViT (Dosovitskiy et al., 2020) offers stronger pre-trained knowledge. Various fine-tuning methods, such as bias tuning (Zaken et al., 2021; Cai et al., 2020), side-tuning (Zhang et al., 2020a), adapter (Rebuffi et al., 2017), and prompt tuning (Jia et al., 2022) have also been studied. Though being able to adjust fewer parameters, these methods still cannot outperform full-network tuning, which is what we focus on.

**Synthetic Data from Generative Models.** Traditional synthetic data are acquired through the renderings from the graphics engines (Dosovitskiy et al., 2015; Peng et al., 2017; Richter et al., 2016). However, this type of synthesis cannot guarantee the quality and diversity of the generated data, resulting in a large gap with real-world data. Recent success in generative models has made synthesizing photo-realistic and high-fidelity images possible, which could be used for training the neural networks for image recognition due to their unlimited generation. For example, early works explored Generative Adversarial Networks (GAN) (Creswell et al., 2018) for image recognition tasks. Besnier et al. (2020) use a class-conditional GAN to train the classifier head and Zhang et al. (2021) uses StyleGAN to generate the labels for object segmentation. Jahanian et al. (2021) adopts the GAN as a generator to synthesize multiple views to conduct contrastive learning. Until recently, the text-to-images models (Dhariwal & Nichol, 2021; Lugmayr et al., 2022; Rombach et al., 2022; Saharia et al., 2022a) have been leveraged to synthesize high-quality data for neural network training due to their effectiveness and efficiency. He et al. (2022) adopts GLIDE (Nichol et al., 2021) to synthesize images for classifier tuning on the CLIP model (Radford et al., 2021). However, simply tuning the classifier may not fully explore the potential of synthetic data. StableRep (Tian et al., 2023) proposes to use Stable Diffusion for generating pre-trained datasets for contrastive learning and leverages the synthetic data from different random seeds as the positive pairs. Azizi et al. (2023) explores synthesizing data under ImageNet label space and finds improved performance. Fill-up (Shin et al., 2023) balances the long-tail distribution of the training data using the synthetic data from text-to-image models.

## 3 PRELIMINARIES

**Problem Setup.** In this paper, our primary focus is exploring a structured pipeline for leveraging synthetic data to enhance transfer learning. Transfer learning addresses the challenge of fine-tuning a pre-trained model, originally trained on a general dataset, so that it performs well on downstream, domain-specific datasets. Rather than solely fine-tuning the classifier in the specific CLIP model (Radford et al., 2021), as outlined by He et al. (2022), we expand our study to incorporate a range of general ImageNet pretrained models while also fine-tuning the entire neural network. Given the massive synthetic data, fine-tuning only the classifier may not fully encode the features of all data into the networks and lead to different findings, which is also discovered in Shin et al. (2023) (See Appendix E). Optimizing the full networks substantially broadens and generalizes our investigative scope, making it a more comprehensive and practical approach to the study.

**Downstream Datasets.** We consider on 10 downstream datasets from different domains, including FGVC Aircraft (Maji et al., 2013), Caltech-101 (Fei-Fei et al., 2004), CUB200-2011 (Wah et al., 2011), Describe Textures Dataset (Cimpoi et al., 2014), Flowers (Nilsback & Zisserman, 2008), Food-101 (Bossard et al., 2014), Pets (Parkhi et al., 2012), SUN397 (Xiao et al., 2010), Stanford Cars (Krause et al., 2013), Stanford Dog (Khosla et al., 2011). The detailed train/val split and the number of classes are shown in the Appendix A.

**Pre-trained Models.** We fine-tune the ResNet-18 model (He et al., 2016) to these downstream datasets to establish the baseline fine-tuning performance for all our explored methods with synthetic data. In Section 4.4, we expand the model to ResNet-50, and Vision Transformers (Dosovitskiy et al., 2020). The training hyper-parameters are detailed in the Appendix A.

**Data Synthesis.** To generate synthetic images, we employ the Stable Diffusion V1.5 (StabilityAI, 2023). All synthesized images are initially created with a resolution of $512 \times 512$ pixels. Subsequently, we adhere to the standard ImageNet preprocessing guidelines (He et al., 2016) to resize these images to $224 \times 224$ pixels during the fine-tuning phase. We utilize the ImageNet text prompt template (Radford et al., 2021) to generate images tailored to specific class names. Detailed information regarding this template is provided in the Appendix B. For instance, for the class

name "Abyssinian" from the Pets dataset, we might randomly select the prompt `"A good photo of the Abyssinian"` for image generation. The Denoising Diffusion Probabilistic Models (DDPM) technique (Ho et al., 2020) is used for sampling the diffusion steps. A guidance scale of 3.5 is applied during the image generation process. Unless otherwise specified, we generate 1,000 images for each class across all datasets. A comprehensive analysis of the factors affecting image synthesis will be studied in Section 4.2 and 4.3. Furthermore, all primary experiments were conducted three times with different random seeds, and the variances across these runs are reported.

# 4    IS SYNTHETIC DATA USEFUL FOR TRANSFER LEARNING?

In the following section, we delve into the effective incorporation of synthetic images in the domain of transfer learning. Initially, we establish that a straightforward amalgamation of real and synthetic images is insufficient for enhancing transfer learning performance. To address this limitation, we propose a framework dubbed bridged transfer, discussed in detail in Sec. 4.1. Within the context of this framework, we further examine the influence of the volume of synthetic images in Sec. 4.2. Additionally, we introduce the concept of dataset style inversion in Sec. 4.3, aimed at refining the stylistic alignment between real and synthetic images, thereby augmenting their utility in transfer learning applications.

## 4.1    HOW TO UTILIZE SYNTHETIC IMAGES FOR TRANSFER LEARNING?

**Transfer Pipelines.** We compare three different transfer learning pipelines with and without synthetic images:

1. **Vanilla Transfer:** This method fine-tunes a model, initially trained on ImageNet, using exclusively real images.
2. **Mixed Transfer:** This method involves fine-tuning the ImageNet pre-trained model with a combination of both real and synthetic images.
3. **Bridged Transfer:** This is a two-stage transfer process. Initially, the ImageNet pre-trained model is fine-tuned solely with synthetic data. Subsequently, in the second stage, it is fine-tuned using only real images.

Figure 2 presents accuracies for three different transfer learning pipelines across 10 downstream datasets. On average, the use of mixed transfer results in a 6–10% reduction in accuracy when compared to vanilla transfer. This suggests that there exists a distributional disparity between synthetic and real images. Employing a naive approach to mix these distributions tends to deteriorate the quality of the training data, consequently leading to suboptimal performance on real test datasets. Conversely, the bridged transfer method demonstrates a notably higher accuracy compared to mixed transfer. This underscores the importance of utilizing synthetic data in a manner that complements rather than obfuscates the real data distribution. In certain instances, the accuracy achieved through bridged transfer surpasses that of vanilla transfer. For instance, the Cars dataset shows an improvement in accuracy by over 6% when using bridged transfer.

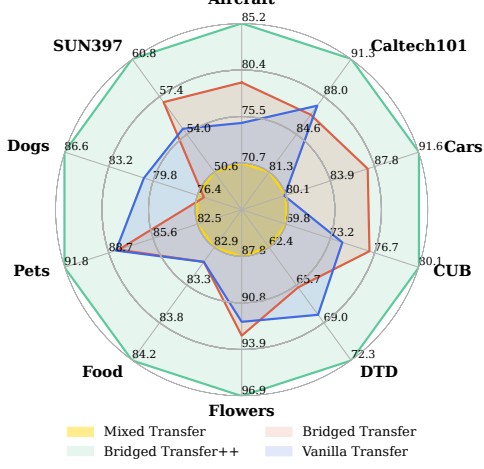

Figure 2: Test accuracy comparison across 10 target tasks using 4 different transfer pipelines.

However, it is crucial to note that bridged transfer does not uniformly yield better performance relative to vanilla transfer across all datasets. To understand the underlying reasons for this variance, we further investigate into the training mechanics of the bridged transfer.

> **Takeaway**: Simple mixing of real and synthetic data fails to improve transfer learning, often resulting in degraded accuracy.

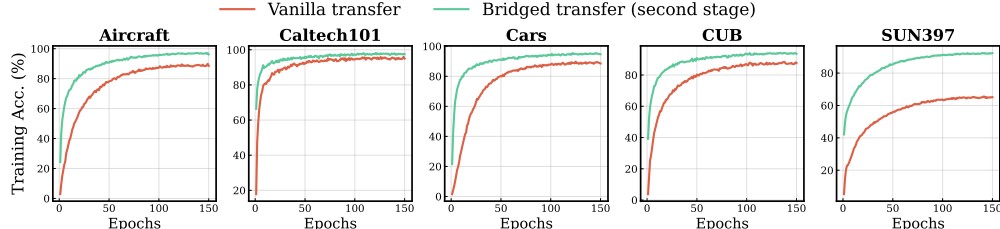

Figure 3: Training accuracy convergence of vanilla and bridged transfer on 5 target datasets. Bridged transfer accelerates the training convergence.

Table 1: Transferability comparison between model pre-trained on ImageNet and model fine-tuned by synthetic images on ten downstream datasets. Our assessment of transferability employs the LEEP score (Nguyen et al., 2020), where a lower magnitude, indicated by less negative numbers, denotes superior transferability (bold).

| Model trained on | Aircraft | Caltech-101 | Cars | CUB | DTD | Flowers | Food | Pets | Dogs | SUN397 |
|---|---|---|---|---|---|---|---|---|---|---|
| **ImageNet** | -4.30 | -4.54 | -5.26 | -5.25 | -3.50 | -4.58 | -1.16 | -3.54 | -4.72 | -5.92 |
| **ImageNet→Synthetic** | **-3.41** | **-3.49** | **-2.20** | **-2.68** | **-1.94** | **-1.93** | **-0.98** | **-1.17** | **-2.36** | **-2.85** |

**Does Synthetic Data Encourage Better Transferability?** While bridged transfer does not consistently outperform vanilla transfer, we provide two pieces of evidence that suggest fine-tuning on synthetic images enhances the transferability of ImageNet pre-trained models to downstream datasets.

Firstly, Figure 3 illustrates the training curves on five downstream training sets (see more datasets in Appendix C), comparing the ImageNet pre-trained model ("Vanilla transfer") to the model fine-tuned on synthetic data ("Bridged transfer"). Across ten datasets, we observe that bridged transfer consistently achieves faster convergence on the real training set as compared to vanilla transfer.

Secondly, we employ the LEEP score (Nguyen et al., 2020) as a quantitative metric to assess the transferability of models to downstream datasets. Table 1 compares the LEEP scores of an ImageNet pre-trained ResNet-18 with a ResNet-18 fine-tuned on synthetic data. The results reveal that a model fine-tuned on synthetic data consistently attains higher transferability against the ImageNet pre-trained model across all the examined downstream datasets.

**Regularization Improves Bridged Transfer.** The aforementioned findings reveal an intriguing phenomenon wherein bridged transfer enhances the transferability of the ImageNet pre-trained model, yet it may not necessarily improve the final test accuracy on downstream tasks. To conduct a detailed examination of the model fine-tuned with synthetic data, we divided it into two distinct components: the classifier (i.e., the last fully-connected layer) and the feature extractor (i.e., all layers preceding the classifier). Our investigation uncovered that the feature extractors of fine-tuned models consistently exhibit better suitability for downstream datasets compared to their pre-trained counterparts, while the same does not hold true for the classifier. For a comprehensive analysis, please refer to the Appendix C. As observed by Li et al. (2020), the last fully connected layer learns more rapidly during transfer learning, suggesting that synthetic data facilitates the feature extractor in learning more diverse features, while the classifier erroneously associates the generated artifacts with class concepts.

Motivated by these insights, we introduce two regularizations aimed at further enhancing the feature extractor and mitigating classifier issues. In the initial stage of bridged transfer, we employ the Mixup loss function (Zhang et al., 2017) to imbue the model with greater discriminative power between classes and a more generalizable feature extractor. Subsequently, after fine-tuning the model on the synthetic dataset, we discard the final linear classifier and reinitialize it randomly before fine-tuning it on downstream real-image datasets, a process referred to as FC Reinit. We name the finalized method bridged transfer++.

We summarize the results in Table 2. With all the regularization techniques introduced here, bridged transfer++ consistently outperforms vanilla transfer. Notably, on several fine-grained datasets, our method exhibits significant improvements. For instance, bridged transfer++ results in an increase

Table 2: Test accuracy of different transfer pipelines on various full-shot downstream datasets.

| Dataset | Aircraft | Caltech-101 | Cars | CUB | DTD | Flowers | Food | Pets | Dogs | SUN397 |
|---|---|---|---|---|---|---|---|---|---|---|
| Vanilla Transfer | 79.7±0.4 | 90.5±0.4 | 83.8±0.1 | 77.6±0.1 | 71.6±0.3 | 95.1±0.0 | 83.5±0.1 | 91.3±0.0 | 83.9±0.1 | 57.9±0.2 |
| Mixed Transfer | 70.7±0.3 | 81.3±0.5 | 80.1±0.4 | 69.8±0.3 | 62.4±0.5 | 87.8±0.1 | 82.9±0.2 | 82.5±0.2 | 76.4±0.4 | 50.6±0.5 |
| Bridged Transfer | 83.9±0.2 | 89.7±0.3 | 90.0±0.2 | 79.7±0.1 | 69.2±0.1 | 96.0±0.0 | 83.5±0.1 | 91.1±0.0 | 79.3±0.2 | 60.3±0.2 |
| + FC Reinit | 84.6±0.5 | 90.7±0.2 | 90.0±0.0 | 79.7±0.0 | 71.7±0.3 | 96.5±0.0 | 83.9±0.1 | 90.2±0.1 | 79.9±0.1 | 60.4±0.2 |
| + FC Reinit & Mixup | **85.2±0.0** | **91.3±0.1** | **91.6±0.1** | **80.1±0.1** | **72.3±0.3** | **96.9±0.1** | **84.2±0.0** | **91.8±0.0** | **86.6±0.2** | **60.8±0.2** |

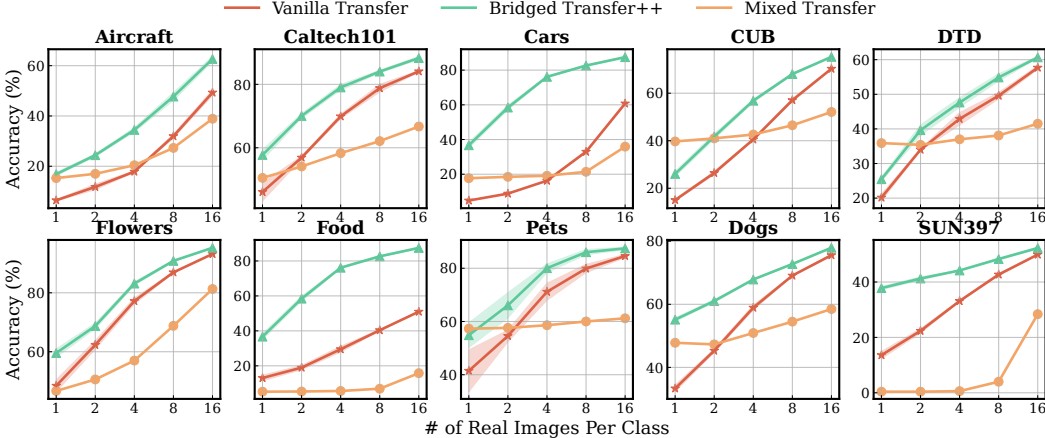

Figure 4: Test accuracy of different transfer pipelines on various few-shot downstream datasets. The x-axis is the number of shots (real images) per class.

of 5.5% and 7.8% in accuracy on the Aircraft and Cars datasets, respectively. These fine-grained datasets are typically challenging to enhance due to difficulties in collecting images and the small differences between classes (Zhang et al., 2020b). A proper way to utilize synthetic data can significantly enhance performance on such datasets.

**Bridged Transfer++ in Few-Shot Regime.** We evaluate bridged transfer++ in a few-shot transfer learning scenario, where each class in a downstream dataset only has $k$ images. Typical values for $k$ include 1, 2, 4, 8, and 16 (He et al., 2022). Bridged transfer++ consistently demonstrates a significant improvement over vanilla transfer in this few-shot setting. Conversely, mixed transfer exhibits a relatively more pronounced accuracy degradation compared to the full-shot setting, as the limited quantity of real images makes it easier to distort the underlying data distribution.

Furthermore, as anticipated, the advantages of synthetic images with bridged transfer++ become even more pronounced compared to the full-shot settings, with improvements of up to 60% observed on the 4-shot Cars dataset scenario.

> **Takeaway**: Fine-tuning the ImageNet pre-trained model on synthetic images enhances its transferability to downstream datasets, resulting in improved transfer learning accuracy across all (full-shot and few-shot) datasets under consideration when appropriate regularizations are applied.

## 4.2 HOW MUCH DATA TO SYNTHESIZE?

In our previous experiments, we generated 1000 synthetic images per class. Now, we delve into the impact of synthetic data volume on the outcomes of bridged transfer++. We conduct an investigation across a spectrum of image quantities per class, ranging from 500 to 3000.

Firstly, we scrutinize the influence of synthetic data volume on full-shot transfer learning, as depicted in Figure 5 (a). The respective numbers of real images per class for the full-shot Aircraft, Cars, and Food datasets are $\{66, 42, 750\}$. Across the range from 500 to 3000 synthetic images per class, it's evident that higher quantities of synthetic images consistently yield greater accuracy, and this improvement has not reached a saturation point yet.

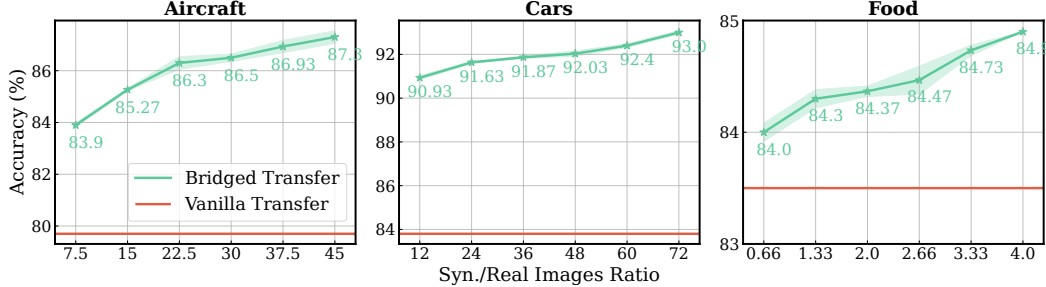

(a) Impact of synthetic data volume in full-shot real data regime. The numbers of real images per class for Aircraft, Cars, and Food datasets are $\{66, 42, 750\}$.

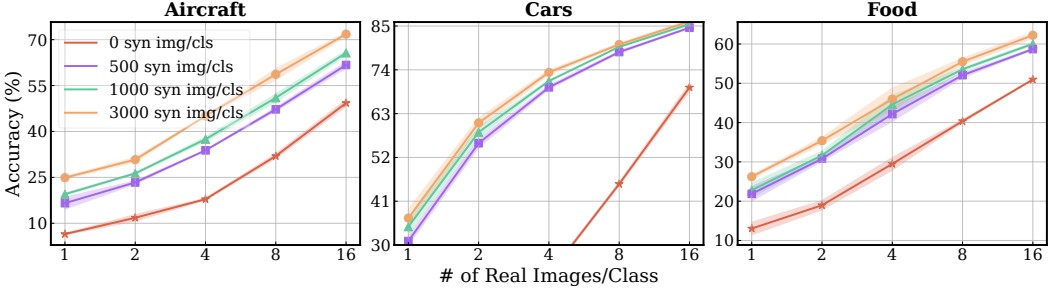

(b) Impact of synthetic data volume in few-shot real data regime.

Figure 5: Investigation on the number of synthetic data used for bridged transfer.

Furthermore, we observe that increasing the number of synthetic images can also yield substantial improvements in few-shot transfer learning performance, as illustrated in Figure 5 (b). For instance, with bridged transfer++, having 3k synthetic images per class results in over a 10% increase in accuracy compared to bridged transfer++ with 500 images per class on the Aircraft dataset.

> **Takeaway**: Increasing the number of synthetic data instances improves the accuracy of Bridged Transfer++, at least within the range of 0.5k to 3k synthetic images/class.

## 4.3 How to Synthesize the Image?

In this section, we examine how various controls on image generation affect the performance of bridged transfer++. In our earlier experiments, we utilized a prompt template with the label name as the text input for the stable diffusion model, along with default hyper-parameters and sampling processes. In this context, we delve into the impact of guidance scale and style token learning for downstream datasets.

**Guidance Scale.** In data synthesis, diversity and fidelity (or text-image alignment) are two key factors affecting the overall quality of the synthetic data. On one hand, the diversity ensures a massive amount of data can be generated for general visual representation learning, on the other hand, the fidelity guarantees the synthetic data is within a similar domain of the real data. The Stable Diffusion uses classifier-free guidance (Ho & Salimans, 2022) to balance the diversity and text-image alignment, which linearly combines text-conditional score estimate $\epsilon(z_t, c)$ and the unconditional score estimate $\epsilon(z_t)$ via the guidance scale w at each step $t$:

$$\tilde{\epsilon}(z_t, c) = w\epsilon(z_t, c) + (1-w)\epsilon(z_t). \tag{1}$$

A higher $w$ encourages higher fidelity but less diversity in the synthetic data. We experiment with the different choices of guidance scale $w$ on 3 datasets: Aircraft, Flowers, Foods by using the guidance scale from $\{2, 3.5, 5, 6.5, 8\}$ and evaluate the bridged transfer performance, as demonstrated in Figure 6. We notice that all choices of GS can improve the transfer performance on three datasets. Generally, we found there is no fixed optimal GS for all datasets, but the impact is relatively modest on bridged transfer++. We empirically found that 3.5 is sufficient to balance diversity and fidelity.

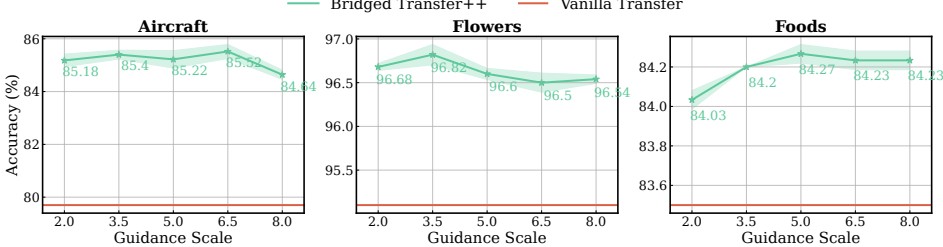

Figure 6: Performance of bridged transfer++ with synthetic data from various guidance scale values.

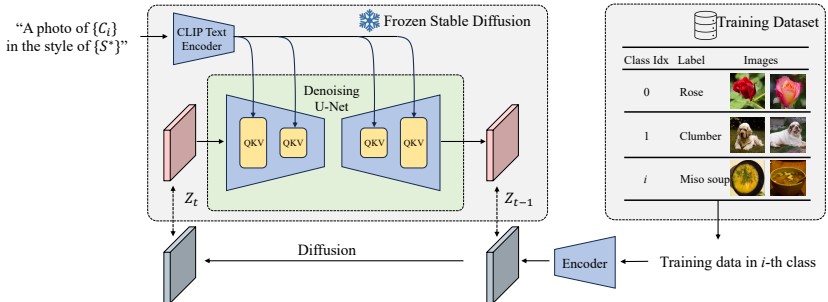

Figure 7: The overview of dataset style inversion. We optimize a single token $S^*$ to represent the style of all real training data.

> **Takeaway**: Bridged transfer++ is robust to the change of guidance scale.

**Dataset Style Inversion.** Aligning the style of synthetic images with that of target real images is a valuable strategy to narrow the distribution gap between synthetic and real data, thereby enhancing the effectiveness of synthetic images for transfer learning. However, articulating the specific style characteristics of a downstream dataset using natural language can be exceedingly challenging.

Inspired by Textual Inversion Gal et al. (2022), we introduce Dataset Style Inversion (DSI), which harnesses the expressive capabilities of a text encoder to define the style of a real image dataset by learning a single style token.

In the DSI framework, our objective is to optimize a solitary token, denoted as $S^*$, to encapsulate the style of the entire dataset. To ensure this style token can generalize effectively across various classes, we employ the prompt "A $C_i$ photo in the style of $S^*$", where $C_i$ represents the label name of the $i$-th class. During the optimization process for $S^*$, we randomly sample a class label in each iteration and map the latent representation of the real images within that class to the denoised latent representation using the following equation:

$$S^* = \arg\min_S \mathbb{E}_{\boldsymbol{z}\sim\mathcal{E}(\boldsymbol{x}),y,\epsilon\sim\mathcal{N}(0,1)} \left[||\epsilon - \epsilon_\theta(\boldsymbol{z}_t, t, C_i(\boldsymbol{y}))||_2^2\right]. \tag{2}$$

This approach effectively encapsulates the style of the entire dataset within a single token. Notably, compared to the original Textual Inversion, our proposed DSI method requires just one-time training for one downstream dataset, resulting in a significant reduction in computational cost. For instance, achieving class-wise textual inversion on the Aircraft dataset, comprising 100 classes, necessitates 5k×100 training iterations hours, while our method takes only 20k training iterations.

To demonstrate the effect of DSI on bridged transfer, we select datasets including Aircraft, Cars, DTD, and SUN397. The results are summarized in the following Table. 3. We noticed that DSI consistently improves the performances on these datasets.

Table 3: Comparison of traditional single template prompt and our DSI in bridged transfer++.

| Synthesis Method | Aircraft | Cars | DTD | Foods | SUN397 |
|---|---|---|---|---|---|
| Single Template | 85.2±0.0 | 91.6±0.1 | 72.3±0.3 | 84.2±0.0 | 60.8±0.2 |
| DSI | **85.8±0.2** | **92.0±0.0** | **72.7±0.3** | **84.6±0.1** | **63.4±0.3** |

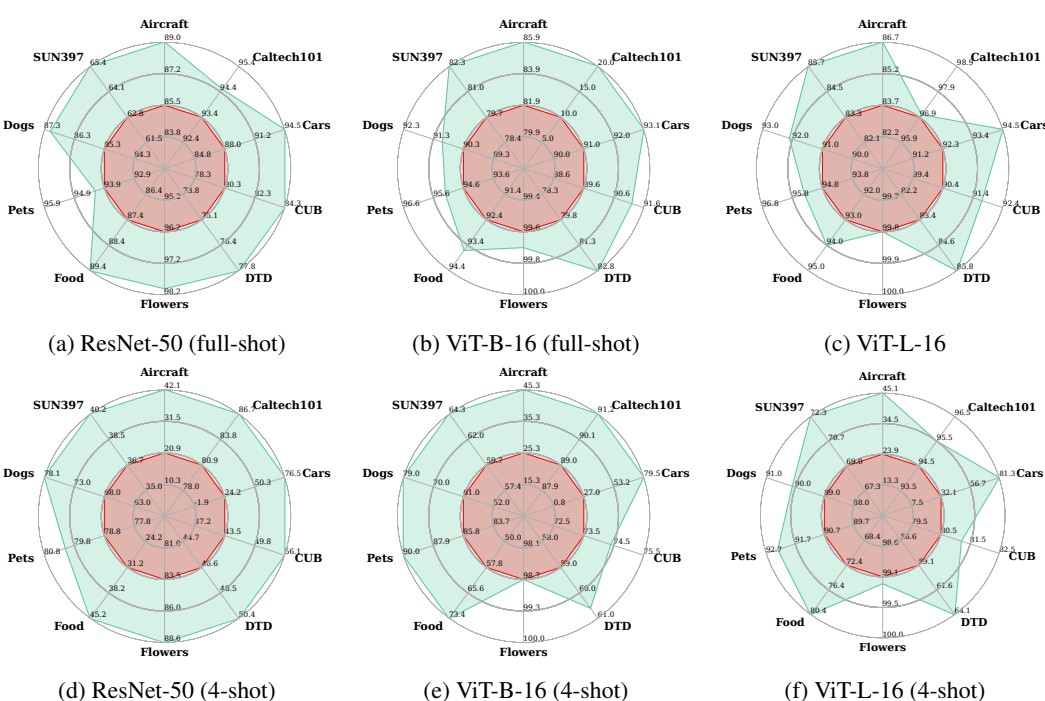

Figure 8: Comparison between bridged transfer++ (green) and vanilla transfer (red) with ResNet-50, ViT-B/L-16 on 10 target datasets. We test the performance under both full-shot and 4-shot regimes.

## 4.4 COMPARISON ON OTHER ARCHITECTURES

In this section, we compare our overall pipeline with vanilla transfer on other network architectures, including ResNet-50 (He et al., 2016) and two variants of Vision Transformers (Dosovitskiy et al., 2020). We implement bridged transfer with 2.5k images per class and apply DSI to synthesize the images for fine-grained datasets like DTD, Flowers, and Caltech101. We test both the full-shot transfer and the 4-shot transfer. Figure 8 demonstrates all the test results. We can find that our method improves the transfer performance in almost all cases. Notably, the most two impact datasets are Aircraft and Cars. As an example, our bridged transfer can improve the 53% accuracy of ViT-B-16 on the Cars dataset. On average, our method significantly improves the few-shot transfer learning results, resulting in 13%, 12%, and 9% accuracy improvement on three architectures, respectively.

## 5 CONCLUSION

This study offers a novel perspective on the utility of synthetic data generated by text-to-image models in the context of ImageNet transfer learning tasks. We meticulously examine three key facets—data utilization, data volume, and data generation control—across multiple downstream datasets. Our innovative two-stage Bridge Transfer framework successfully addresses the limitations found in naively blending synthetic and real data, providing a more effective and nuanced approach to improve transferability and training convergence. The empirical evidence demonstrates consistent performance enhancements, further reinforced by our Dataset Style Inversion technique, which effectively aligns the stylistic characteristics of synthetic and real images.

The findings have profound implications for the field of computer vision, particularly for applications that not have access to abundant labeled data in the target domain. The approach presented not only shows the potential to substantially reduce the data requirements for effective transfer learning but also offers a pathway to leverage the burgeoning capabilities of Generative AI.

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

# A  EXPERIMENTAL DETAILS

In this section, we provide the experimental details. First, Table 5 below shows the detailed number of classes and the train/validation split of each dataset we used in the main paper.

Table 4: Classification datasets used in this paper.

| Dataset | #Classes | Size (Train/Val) | Acc. Metric |
|---|---|---|---|
| Aircraft (Maji et al., 2013) | 100 | 6,667/3,333 | Mean Per-Class |
| Caltech101 (Fei-Fei et al., 2004) | 101 | 3,030/5,647 | Mean Per-Class |
| Cars (Krause et al., 2013) | 120 | 8,144/8,041 | Top-1 |
| CUB-200 (Wah et al., 2011) | 200 | 5,994/5,794 | Top-1 |
| DTD (Cimpoi et al., 2014) | 47 | 3,760/1,880 | Top-1 |
| Dogs (Khosla et al., 2011) | 120 | 12,000/8,580 | Top-1 |
| Flowers (Nilsback & Zisserman, 2008) | 102 | 2,040/6,149 | Mean Per-Class |
| Food (Bossard et al., 2014) | 101 | 75,750/25,250 | Top-1 |
| Pets (Parkhi et al., 2012) | 37 | 3,680/3,669 | Mean Per-Class |
| SUN397 (Xiao et al., 2010) | 397 | 19,850/19,850 | Top-1 |

Here we show the training hyper-parameters we used for ResNet-18 fine-tuning. For all experiments, we use the SGD optimizer followed by a cosine annealing decay schedule. We fix the weight decay to $5e - 4$, the batch size to 64, and the number of training epochs to 150. The only flexibly hyper-parameter is the learning rate, which was chosen from $\{0.1, 0.03, 0.01, 0.003, 0.001\}$ for maximum performance. In practice, the transfer performance can be further increased if we conduct a grid search on other hyper-parameters as well.

Table 5: Training hyper-parameters of ResNet-18.

| Dataset | LR | Weight decay | Batch size | Epochs |
|---|---|---|---|---|
| Aircraft (Maji et al., 2013) | 0.1 | $5e - 4$ | 64 | 150 |
| Caltech101 (Fei-Fei et al., 2004) | 0.003 | $5e - 4$ | 64 | 150 |
| Cars (Krause et al., 2013) | 0.01 | $5e - 4$ | 64 | 150 |
| CUB-200 (Wah et al., 2011) | 0.01 | $5e - 4$ | 64 | 150 |
| DTD (Cimpoi et al., 2014) | 0.003 | $5e - 4$ | 64 | 150 |
| Dogs (Khosla et al., 2011) | 0.01 | $5e - 4$ | 64 | 150 |
| Flowers (Nilsback & Zisserman, 2008) | 0.01 | $5e - 4$ | 64 | 150 |
| Food (Bossard et al., 2014) | 0.01 | $5e - 4$ | 64 | 150 |
| Pets (Parkhi et al., 2012) | 0.003 | $5e - 4$ | 64 | 150 |
| SUN397 (Xiao et al., 2010) | 0.003 | $5e - 4$ | 64 | 150 |

For few-shot learning, we set training epochs to 100 and keep weight decay & batch size the same as the full-shot learning scenario. The learning rate is also chosen from the set for maximized performance as aforementioned.

As for other architectures, the hyper-parameter search policy of ResNet-50 is kept the same with ResNet-18. As for the ViTs, we set the weight decay to 0 and the batch size to 128. The training epochs as well as the learning rate search are the same with ResNets.

# B  PROMPT TEMPLATE

We use the following template (adopted in Radford et al. (2021)) to generate the prompt for the label-based single template data. $C*$ is the label name.

1. `a photo of a ` $C*$ `.`
2. `a rendering of a ` $C*$ `.`

3.  a cropped photo of the $C*$.
4.  the photo of a $C*$.
5.  a photo of a clean $C*$.
6.  a photo of a dirty $C*$.
7.  a dark photo of the $C*$.
8.  a photo of my $C*$.
9.  a photo of the cool $C*$.
10. a close-up photo of a $C*$.
11. a bright photo of the $C*$.
12. a cropped photo of a $C*$.
13. a photo of the $C*$.
14. a good photo of the $C*$.
15. a photo of one $C*$.
16. a close-up photo of the $C*$.
17. a rendition of the $C*$.
18. a photo of the clean $C*$.
19. a rendition of a $C*$.
20. a photo of a nice $C*$.
21. a good photo of a $C*$.
22. a photo of the nice $C*$.
23. a photo of the small $C*$.
24. a photo of the weird $C*$.
25. a photo of the large $C*$.
26. a photo of a cool $C*$.
27. a photo of a small $C*$.

## C  ADDITIONAL EXPERIMENTAL RESULTS

In this section, we provide the additional experimental results mentioned in our paper. In Figure 9, we show the rest 5 datasets' convergence. The finding aligns well with our former observation.

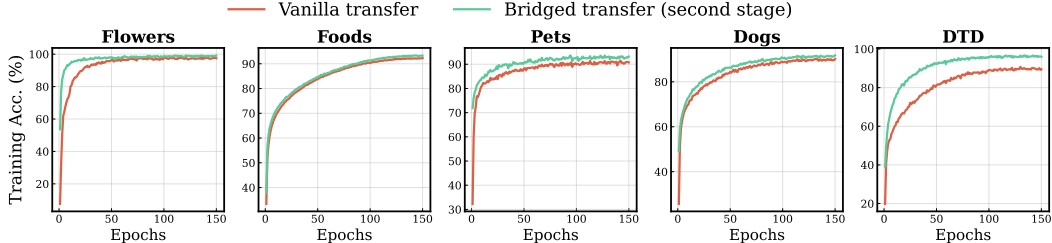

Figure 9: Training accuracy convergence of vanilla and bridged transfer on the other 5 datasets.

Next, we conduct experiments on fixed-feature transfer to study the effect of the classifier head during bridged transfer. In particular, we freeze the feature extractor and only optimize the FC layer. We compare three cases: (1) vanilla transfer where we optimize the FC of the ImageNet pretrained model, (2) bridged transfer where we optimize the FC of the synthetic data fine-tuned model, and (3) bridged transfer with reinitialized FC layer.

We test these experiments on Flowers, Dogs, and DTD datasets. With full-network transfer, the bridged transfer has lower performance on these 3 datasets. The results are shown in Table 6. It can

Table 6: Comparison of different transfer pipelines evaluated with fixed-feature transfer.

| Synthesis Method | Flowers | Dogs | DTD |
|---|---|---|---|
| Vanilla Transfer | 86.5 | 84.1 | 63.2 |
| Bridged Transfer | 84.4 | 75.4 | 65.1 |
| Bridged Transfer + FC Reinit | **91.9** | **84.5** | **66.2** |

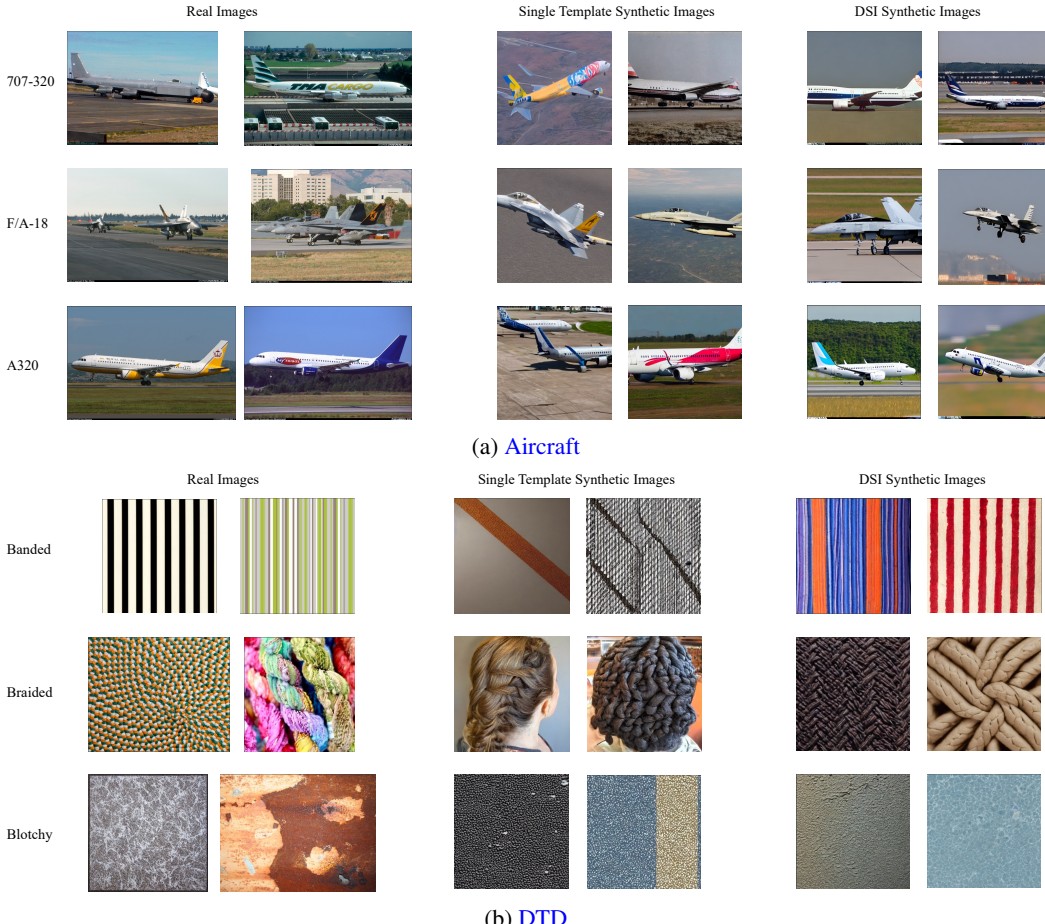

(a) Aircraft

(b) DTD

Figure 10: Example images generated from Single Template and our Dataset Style Inversion.

be observed that the bridged transfer has much lower accuracy than the vanilla transfer. However, if we keep the feature extractor fine-tuned with the synthetic data and only optimize the FC layer, namely brigded transfer with FC reinitialization, the performance is even better than vanilla transfer. This finding aligns with our statement that bridged transfer increases transferability, especially the feature extractor of the network.

# D  QUALITATIVE EVALUATION

In this section, we provide some example images of original training data and synthetic data. For synthetic data, we additionally generate it with Single Template and DSI. Figure 10 shows the visualization on DTD and Aircraft datasets. We can observe that DSI generates images that are visually closer to real images than the Single Template method. For example, the *Braided* class on DTD

dataset tends to generate human braids rather than actual texture with Single Template, while DSI can correctly generate the corresponding texture.

# E DISCUSSION WITH HE ET AL. (2022)

He et al. (2022) also experiments with the effect of synthetic data during transfer learning. Specifically, they test the zero-shot and few-shot transfer learning performance with CLIP classifier tuning (CT) (Wortsman et al., 2022), which only changes the last layer of the CLIP model (Radford et al., 2021). Instead, our method focuses on the whole network finetuning given an ImageNet pretrained model. Both methods propose to mitigate the gap between synthetic data and the real data. He et al. (2022) uses Real Guidance that replaces the first several denoising steps with the noisy latent from the real images, while our method tries to invert a token embedding to directly characterize the dataset style.

Here, we also experiment with this setup using our synthesis and Bridged Transfer pipeline. We conduct our experiments on the Aircraft and the Pets datasets. 1000 images per class are generated. We also employ the Real Guidance (RG) strategy (He et al., 2022). The results are summarized in the Table below.

Table 7: Transfer performance of CLIP classifier tuning.

| Transfer Method | Synthesis Method | Aircraft | | Pets | |
|---|---|---|---|---|---|
| | | 1-shot | 8-shot | 1-shot | 8-shot |
| Vanilla Transfer | No Synthesis | 20.22 | 32.70 | 86.21 | 87.89 |
| Mixed Transfer | Single Template | 21.90 | 31.83 | 87.08 | 87.79 |
| Bridged Transfer | Single Template | 20.85 | 31.71 | 87.46 | 87.76 |
| Mixed Transfer | Real Guidance | 22.62 | 33.40 | 87.48 | 89.45 |
| Bridged Transfer | Real Guidance | 21.77 | 32.84 | 87.33 | 89.33 |

Based on the results, we made several observations.

(1) The Real Guidance synthesis can improve the transfer learning performance when compared to the Single Template, which is also discovered in He et al. (2022).

(2) For Single Template image synthesis, we found that under the 8-shot transfer, it cannot improve the vanilla transfer performance, this is similar to our finding that, unless guided by real data, regular synthetic data has a distribution shift and cost accuracy decreases in mixed transfer.

(3) Different from our work, the bridged transfer does not always perform better than the mixed transfer using CLIP CT, also discovered in He et al. (2022) Table 7. We think the reasons are twofold: First, in our paper, we found that synthetic data may lead to bias in the final classifier layer, therefore, we employ the FC reinitialization in Bridged Transfer++. In He et al. (2022), since only the classifier is allowed to be tuned, it is not surprising that bridged transfer may have lower performance than mixed transfer when only optimizing the classifier. Second, He et al. (2022) uses different batch sizes to balance the synthetic data and real data (512 for synthetic data and 32 for real data) so that the network weighs more in real data to prevent distribution shifts.

