# OpenReview forum: "IS SYNTHETIC DATA USEFUL FOR TRANSFER LEARNING? AN INVESTIGATION INTO DATA GENERATION, VOLUME, AND UTILIZATION"
_ICLR.cc/2024/Conference — Submitted to ICLR 2024_

### Official Review · Reviewer_yiMQ · 2023-10-27

**Soundness:** 1 poor
**Presentation:** 2 fair
**Contribution:** 2 fair
**Rating:** 5
**Confidence:** 4

**Summary:**

The paper proposes using a large-scale pretrained generative image model to improve transfer learning. To do so, the method proposes querying for images based on the label to retrieve similar images to those in downstream tasks. Then 1) first fine-tunes a pretrained image model (on Imagenet) and then 2) further fine-tunes it using real images from the downstream task. The paper shows that this technique, called Bridge transfer learning, improves over directly fine-tuning on Imagenet (vanilla transfer) and fine-tuning with a mix of synthetic and real images (mixed transfer). The paper also proposes fine-tuning the diffusion model to make it generate images in the style of a given dataset, which improves performance and does not require per-class finetuning.

**Strengths:**

- The method is simple and may have practical value when a data source is only accessible through a pretrained generative model.

- The method is tested using fine-grained classification tasks, which may be hard for generally pretrained models to perform well.

- The study in 4.3 about inverting the dataset style is interesting and has practical utility, as it is more efficient than fine-tuning the diffusion model for every class.

**Weaknesses:**

- The main issue with the paper is that it is comparing:

1) **Baseline**: a training procedure that only takes *Imagenet-1k* data (1.3M images) + downstream data
2) **Method Proposed**: a training procedure that takes *Imagenet-1k* + *LAION-5B* (StableDiffusion).

The proposed technique uses Imagenet-1k + LAION-5B (distilled into StableDiffusion), so it should only be compared against methods that use the same or similar amounts of *data* and *compute*. For example, how does a model pretrained on LAION-2/5B (like OpenCLIP, readily available online in [1]) perform when finetuned or even zero-shot? It is unsurprising that a machine learning method that has access to 5B images (with fine-grained labels) performs better than one that only has access to 1.3M images (with 1k discrete annotations).

Because of the above, the paper fails to answer the question of whether it is better to use large-scale image data to first train for *classification* (i.e. OpenCLIP) and then finetune on small data, or use large-scale image data to first train a *generative* image model, which can generate synthetic data to augment the small-data (as proposed). This is an important (and easy to test) question, that the paper does not resolve because of the choice of using a pretrained model on Imagenet-1k and not on the full LAION.

Furthermore, the baseline also uses less overall compute than the method proposed, as training on Imagenet-1k takes an order of magnitude less than training StableDiffusion.

- Section 4.3 (effect of guidance scale) is of moderate interest, since it may depend on the generative model. There are simple studies that are more general and would give more insights into whether it is worth doing transfer learning with generative models. For example, is it better to train with all "Aircraft" images from LAION-X (possibly without fine-grained labels) and then finetune, or to get fine-grained images from StableDiffusion (different types of aircraft models, which may be hard to query for on LAION-X) and then finetune?

- What are the statistics of the data trained on StableDiffusion with respect to the downstream tasks? For example, for the class Aircraft-Boeing 787 or for a specific car model, it may be the case that LAION-X has more than 3k examples. If one cannot generate more than 3k examples (as in Figure 5, because of computational constraints?), why not use the data directly available? Looking at this and pointing out clear examples where one can generate much more quantity/diversity data than that present in LAION-X would be a good insight. For example, is there a specific type of Aircraft model that is hardly present in LAION-X, but one can generate an infinite amount of realistic images for it?

- There may be dataset contamination, which is not discussed, i.e., the test samples of the downstream tasks may be part of LAION-X, while this is not the case for the baseline. Does the model improve performance when prompted with "generate a * with the style of LSUN" without dataset adaptation?

- Visual examples would make the paper stronger. For example, when sampling the different "aircraft", how different are the images from the downstream task? How do these images change when doing dataset Style Inversion?

- Additionally, evaluating on tasks that are speciallized (like medical, satellite imaging...) that may not be as prevalent on LAION-X as "natural" images (vehicles, animals...) would make for a stronger paper and ameliorate some of the concerns expressed in the points above.

**Questions:**

- What is the percentage of images from synthetic and real on the Mixed transfer experiment?
- In LEEP analysis in Table 1, isn't it expected that a model that has lower train/test accuracy (as seen in Figure1 for imagenet pretrained vs bridged pertaining) performs worse on this metric too? Is the difference in transferability (when tested with this metric) also present when two models compared have the same training loss?
- See weaknesses

---

> ### Author Response · Authors · 2023-11-21
> **Reply to Reviewer yiMQ (Part 1)**
>
> Thank you for your comments.
>
> Q0: Involved data source
>
> A0: Thank you for raising this intriguing point. We concur with the reviewer that our "Baseline and Proposed Method involve different data sources." Nonetheless, we would like to clarify that our experiments reveal that "mixed transfer" (employing both ImageNet-1k and LAION-5B) often falls short compared to "vanilla transfer" (solely ImageNet-1k). This outcome suggests that the mere expansion of data does not guarantee enhanced performance. To explicitly reiterate the primary goals of this research: 1) We delve into how **off-the-shelf** stable diffusion can unlock new avenues for boosting transfer learning efficacy. 2) We examine the strategic leverage of stable diffusion through our novel "bridged transfer" and "dataset style inversion" techniques.
>
> Furthermore, we believe it is essential to highlight that our study contributes to the ongoing efforts of integrating text-to-image models into established deep learning workflows. For instance, [1] harnesses synthetic images from Stable Diffusion in place of the CC3M dataset, thereby assessing the impact on model performance with synthetic against original CC3M images; [2] employs Stable Diffusion to enrich the ImageNet 1K dataset, leading to performance enhancements against original ImageNet 1K.
>
> Q1: Because of the above, the paper fails to answer the question of whether it is better to use large-scale image data to first train for classification (i.e. OpenCLIP) and then finetune on small data, or use large-scale image data to first train a generative image model, which can generate synthetic data to augment the small-data (as proposed). This is an important (and easy to test) question, that the paper does not resolve because of the choice of using a pretrained model on Imagenet-1k and not on the full LAION.
>
> A1: We ran the experiments based on the reviewer’s suggestion. Note that CLIP-based tuning is different from traditional transfer learning. For example, for the CLIP-based model, researchers found that classifier finetuning outperforms full network finetuning [3], while for normal CNNs, full network finetuning works better [4]. We compare the performance of CLIP-ResNet50 with Classifier Tuning [3]  and ImageNet-ResNet50 with Bridged Transfer++ on the Aircraft and Pets dataset. Here are the results with 1/8/full-shot data regime.
>
> | Model                             | Aircraft 1-shot | Aircraft 8-shot | Aircraft Full-shot | Pets 1-shot | Pets 8-shot | Pets Full-shot |
> |-----------------------------------|-----------------|-----------------|--------------------|-------------|-------------|----------------|
> | CLIP-ResNet50 with CT                    | 20.22           | 32.70           | 44.50              | 86.21       | 87.89       | 89.0           |
> | ImageNet-ResNet50 with Bridged Transfer ++ | 26.0            | 60.64           | 89.0               | 79.5        | 88.61       | 93.5           |
>
> As demonstrated in the table, fine-tuning CLIP generally yields lower accuracy compared to Bridged Transfer++. There are several reasons for this: (1) CLIP's fine-tuning concentrates on the prompt and classifier, whereas full network fine-tuning enables the model to enhance both the feature extractor and classifier using synthetic data for downstream tasks. (2) Bridged Transfer++ gains greater advantages from an increased synthesis of images as discussed in Section 4.2, whereas CT's performance largely depends on the pre-trained model weights and a constrained set of real images.
>
>
> [1] Tian, Yonglong, Lijie Fan, Phillip Isola, Huiwen Chang, and Dilip Krishnan. "StableRep: Synthetic Images from Text-to-Image Models Make Strong Visual Representation Learners." NeurIPS 2023.
>
> [2] Azizi, Shekoofeh, Simon Kornblith, Chitwan Saharia, Mohammad Norouzi, and David J. Fleet. "Synthetic data from diffusion models improves imagenet classification." arXiv preprint arXiv:2304.08466 (2023).
>
> [3] Wortsman, Mitchell, Gabriel Ilharco, Jong Wook Kim, Mike Li, Simon Kornblith, Rebecca Roelofs, Raphael Gontijo Lopes et al. "Robust fine-tuning of zero-shot models." In Proceedings of the IEEE/CVF Conference on Computer Vision and Pattern Recognition, pp. 7959-7971. 2022.
>
> [4] Shin, Joonghyuk, Minguk Kang, and Jaesik Park. "Fill-Up: Balancing Long-Tailed Data with Generative Models." arXiv preprint arXiv:2306.07200 (2023).

---

> ### Author Response · Authors · 2023-11-21
> **Reply to Reviewer yiMQ (Part 2)**
>
> Q2: Section 4.3 (effect of guidance scale) is of moderate interest, since it may depend on the generative model. There are simple studies that are more general and would give more insights into whether it is worth doing transfer learning with generative models. For example, is it better to train with all "Aircraft" images from LAION-X (possibly without fine-grained labels) and then finetune, or to get fine-grained images from StableDiffusion (different types of aircraft models, which may be hard to query for on LAION-X) and then finetune?
>
> Q3: What are the statistics of the data trained on StableDiffusion with respect to the downstream tasks? For example, for the class Aircraft-Boeing 787 or for a specific car model, it may be the case that LAION-X has more than 3k examples. If one cannot generate more than 3k examples (as in Figure 5, because of computational constraints?), why not use the data directly available? Looking at this and pointing out clear examples where one can generate much more quantity/diversity data than that present in LAION-X would be a good insight. For example, is there a specific type of Aircraft model that is hardly present in LAION-X, but one can generate an infinite amount of realistic images for it?
>
> A2/3: Thanks for your questions. To the best of our knowledge, the reviewer is asking the question of whether using Stable Diffusion data to perform is worth it or not when one can access some real data (for a specific class). To answer this question, we provide two perspectives.
>
> 1. Would LAION-X images help for target dataset transfer learning? We do not verify this question since LAION-X does not give accurate labels for each image. Rather, each image is paired with a caption. For example, when we retrieve the images of class “747-200 Airplan” from here (https://rom1504.github.io/clip-retrieval/?back=https%3A%2F%2Fknn.laion.ai&index=laion5B-L-14&useMclip=false&query=747-200+Airplane), lots of different types of airplane have been retrieved in this case. Therefore, it is hard to determine how to adapt LAION-X images into the supervised training pipeline right now. And this is beyond the scope of our paper.
>
> 2. To quantitatively convert the effect of synthetic data to the effect of real data, we can study this question in few-shot learning scenarios. We run the vanilla transfer from 1-shot to 50-shot one by one and match the accuracy of the bridged transfer. Therefore, we can measure additional samples by evaluating how much more data the vanilla transfer would need to realize the same accuracy as the bridged transfer. The results for the Aircraft dataset of ResNet-18 are shown below:
>
> | Accuracy | Bridged Transfer (w. 3k images / cls) | Vanilla Transfer |
> |----------|---------------------------------------|------------------|
> | 25.1     | 1-shot                                | 6-shot           |
> | 45.0     | 4-shot                                | 14-shot          |
> | 58.8     | 8-shot                                | 24-shot          |
> | 71.4     | 16-shot                               | 40-shot          |
>
> It can be found that synthetic data can effectively improve the performance of vanilla transfer and equal to using more images. In reality, collecting the real data may have some other issues like costs and privacy, please see our reply to Reviewer HV3t.
>
> Q4: There may be dataset contamination, which is not discussed, i.e., the test samples of the downstream tasks may be part of LAION-X, while this is not the case for the baseline. Does the model improve performance when prompted with "generate a * with the style of LSUN" without dataset adaptation?
>
> A4: Thanks for your suggestion. We experimented with this prompt and rerun it on the Aircraft dataset, there was no explicit performance improvement with this prompt (within 0.1% accuracy difference).
>
>
> Q5: Visual examples would make the paper stronger. For example, when sampling the different "aircraft", how different are the images from the downstream task? How do these images change when doing dataset Style Inversion?
>
> A5: Thank you for your advice, we have added visualizations in Appendix D in our revised version, please check it.
>
> Q6: Additionally, evaluating tasks that are specialized (like medical, satellite imaging...) that may not be as prevalent on LAION-X as "natural" images (vehicles, animals...) would make for a stronger paper and ameliorate some of the concerns expressed in the points above.
>
> A6: Thank you for your suggestions. We conducted EuroSAT image classification under the few-shot learning regime (since the full-shot test performance is nearly 100%), and we synthesized 1000 images per class for this dataset. Here are the results.
>
> | Method             | 1-shot | 2-shot | 4-shot | 8-shot | 16-shot |
> |--------------------|--------|--------|--------|--------|---------|
> | Vanilla Transfer   | 49.1   | 60.5   | 65.5   | 78.0   | 85.2    |
> | Bridged Transfer++ | 64.2   | 66.7   | 72.2   | 79.9   | 86.8    |

---

> ### Author Response · Authors · 2023-11-21
> **Reply to Reviewer yiMQ (Part 3)**
>
> Q7: What is the percentage of images from synthetic and real on the Mixed transfer experiment?
>
> A7: In the mixed transfer experiments, we synthesized 1000 images per class. As for the real images per class, we refer you to Appendix A, Table 4 for a detailed description. Roughly speaking, the number of real images per class is within the range of 30~750.
>
> Q8: In LEEP analysis in Table 1, isn't it expected that a model that has lower train/test accuracy (as seen in Figure1 for imagenet pretrained vs bridged pertaining) performs worse on this metric too? Is the difference in transferability (when tested with this metric) also present when two models compared have the same training loss?
>
> A8: The higher the LEEP score, the better the transferability. However, we want to emphasize that the LEEP score is evaluated without any finetuning, instead, it only involves forwarding the target dataset to the pre-trained model. Therefore, our Table 1 and Figure 1 results are the same.

---

> > ### Comment · Reviewer_yiMQ · 2023-11-22
> > **Thanks for the clarifications**
> >
> > Thanks for the detailed rebuttal. The works cited in the rebuttal [1, 2] go beyond the training data available (for the image generator) when testing on the evaluation tasks. For example, [2] samples more visual-text pairs than available in the original datasets, showing that with this extended dataset, they can improve performance with respect to the pretraining dataset. On the other hand, [1] is more similar to the proposed approach but still samples a large collection of samples 1.3M*3 for 1k classes, so it is less likely that one can construct an effective dataset from the original used to pretrain the diffusion model. In the case studied in the paper, the evaluation tasks are much smaller scale (~10k samples), so the gains may likely come from the usage of a billion-scale dataset and not the generative sampling.
> > Still, it is true that 1) it may not be trivial to query for the best images from the original dataset (although reasonable baselines should be tested) or use pretrained classification models on it and 2) when sampling from pretrained generative models, the method proposed is better than vanilla sampling and training. With this, the paper proposes an effective approach to use large-scale datasets for low data-regime (in the form of having this compressed into a generative model), so I have downgraded the importance of this issue.
> >
> > A2) Despite querying will produce false positives/negatives, noisy training may be better than the proposed approach, and I think it should be tested.
> >
> > The CLIP results are contradictory (why does Pets 1-shot perform great, Aircraft bad? Does this happen for multiple datasets?) and point out that training methodology in [3] may not be good for this setting. Why does Imagenet-1k + finetune on full Aircraft gets 79.7 (vanilla training in Table 1), but when using CLIP, performance is 44.50? The proper comparison should just replace the Imagenet-1k Resnet-50 for the CLIP Resnet-50, both using Vanilla transfer (do full fine-tuning, everything else being the same), as it may be hard to setup the hyperparameters for [3]. How does zero-shot CLIP perform (which does not need training) and why hasn't it been tested?
> >
> > Thanks for Figure 10, as it is appealing in showcasing the DSI technique. I would encourage you to move it to the main paper.
> >
> > Although some concerns have not been resolved, I've increased my score and hope the authors will include the reviewer's discussion in the final manuscript.

---

> > > ### Author Response · Authors · 2023-11-23
> > > **Further Reply to Reviewer yiMQ (Part 1)**
> > >
> > > Thanks for replying to our rebuttal and increasing the score. We’d like to reply to your comments in detail.
> > >
> > > Q1: In the case studied in the paper, the evaluation tasks are much smaller scale (~10k samples), so the gains may likely come from the usage of a billion-scale dataset and not the generative sampling.
> > >
> > > Q2: Despite querying will produce false positives/negatives, noisy training may be better than the proposed approach, and I think it should be tested.
> > >
> > > A1/2: We would like to clarify the intent and constraints of our study: it specifically considers scenarios where one has access to the off-the-shelf Stable Diffusion model, but lacks the ability to directly utilize the original LAION-X dataset. Our focus is on investigating how synthetic imagery generated by Stable Diffusion can facilitate transfer learning in practical applications.
> > >
> > > Although employing the LAION-X dataset directly—for instance, by querying samples from it or generating labels from captions—is an intriguing and forward-looking research topic, it also entails significant storage, processing, labeling, and additional computational resources. This could potentially yield superior results, but not without introducing substantial effort. Besides, how to construct a noisy dataset from LAION-X for supervised pre-training remains an open problem at this time.
> > >
> > > Our methodology, which leverages Stable Diffusion, represents a departure from the direct utilization of LAION-X. It is critical to understand that if our technique is to be considered as 'indirectly employing' the LAION-X dataset, it is with the specific intent of enhancing the efficacy of this indirect usage since simply mixing synthetic images with real ones does not work. While a comparative analysis between 'direct' and 'indirect' employment of LAION-X might indeed yield intriguing insights, such an investigation falls beyond the scope of our current research objectives.
> > >
> > > In response to ‘the evaluation tasks are much smaller scale (~10k samples), so the gains may likely come from the usage of a billion-scale dataset and not the generative sampling.’, we respectfully clarify two critical points: Firstly, our methodology did not entail direct utilization of such a large-scale dataset. Instead, we employed the pre-trained Stable Diffusion model to generate synthetic images as outlined in our work. The experimental results clearly indicate that our proposed bridged transfer framework and novel generation technique, dubbed 'dataset style inversion,' are responsible for the performance gains. Secondly, as detailed in Section 4.2 of our paper, we demonstrate a clear positive correlation between the increase in the number of synthetic images and improved task performance, and a saturation effect remains elusive.
> > >
> > > To reiterate, our research does not attempt to compare the efficacy of using Stable Diffusion versus the direct exploitation of the LAION-X dataset. Our study centers on examining efficacious strategies for capitalizing on readily available models like Stable Diffusion to bolster transfer learning, particularly in scenarios characterized by limited data.

---

> > > ### Author Response · Authors · 2023-11-23
> > > **Further Reply to Reviewer yiMQ (Part 2)**
> > >
> > > Q3: The CLIP results are contradictory (why does Pets 1-shot perform great, Aircraft bad? Does this happen for multiple datasets?) and point out that training methodology in [3] may not be good for this setting. Why does Imagenet-1k + finetune on full Aircraft gets 79.7 (vanilla training in Table 1), but when using CLIP, performance is 44.50? The proper comparison should just replace the Imagenet-1k Resnet-50 for the CLIP Resnet-50, both using Vanilla transfer (do full fine-tuning, everything else being the same), as it may be hard to setup the hyperparameters for [3]. How does zero-shot CLIP perform (which does not need training) and why hasn't it been tested?
> > >
> > > A3: Thanks for your question. We would like to add more clarifications. The CLIP model is known for its extraordinary zero-shot and few-shot performance. The original paper [4] even discovers that on some datasets the linear probing performance is inferior to the language-guided zero-shot performance. Meanwhile, neither the original CLIP [4] nor other follow-up works try to perform end-to-end whole network finetuning, as CLIP does not have an advantage in this. We guess this is because it's trained by contrastive less and most contrastive models are only good at Linear Probing. Instead, tuning the classifier or the prompt of the CLIP may bring more advantages in the transfer learning.
> > >
> > > Given the different pre-training loss functions, finetuning, and inference mechanisms between CLIP and ImageNet models, almost no work has been done to systematically compare their performance apples to apples. On some general tasks datasets, CLIP could have better few-shot performance than ImageNe transfer, while on some fine-grained datasets like Aircraft or under full-shot data regime, ImageNet End2End finetuning is better. Here, we test the performance on 4 full-shot datasets (Aircraft, Pets, Cars, DTD) with CLIP-based ResNet50 and ImageNet-based ResNet-50. For CLIP-based ResNet-50, we experimented with end-to-end finetuning using the same learning hyperparameters.
> > >
> > > | Model      | Transfer Method               | Aircraft | Pets | Cars | DTD  |
> > > |------------|-------------------------------|----------|------|------|------|
> > > | CLIP-R50   | No FT, Zero-shot              | 16.8     | 85.7 | 54.5 | 42.5 |
> > > | CLIP-R50   | Linear Probing                | 39.5     | 76.4 | 70.8 | 68.9 |
> > > | CLIP-R50   | Classifier Tuning [3]         | 44.5     | 89.0 | 79.3 | 72.6 |
> > > | CLIP-R50   | End2End FT (Vanilla Transfer) | 60.0     | 80.8 | 38.6 | 54.7 |
> > > | ImgNet-R50 | Vanilla Transfer              | 85.5     | 93.3 | 88.0 | 75.1 |
> > > | ImgNet-R50 | Bridged Transfer++            | 89.0     | 93.5 | 94.5 | 77.8 |
> > >
> > >
> > > Based on the results, we can find that CLIP-based ResNet-50 does not benefit from the end-to-end finetuning. As aforementioned, it is hard to compare CLIP and ImageNet transfer performance as they have their own advantages.
> > >
> > >
> > > Continued Reference
> > >
> > > [4] Radford, Alec, Jong Wook Kim, Chris Hallacy, Aditya Ramesh, Gabriel Goh, Sandhini Agarwal, Girish Sastry et al. "Learning transferable visual models from natural language supervision." In International conference on machine learning, pp. 8748-8763. PMLR, 2021.

---

> ### Comment · Reviewer_yiMQ · 2023-11-23
> **Thanks for the clarification**
>
> I will take into account the additional comments when reaching a consensus on the reviewer discussion period.
>
> Still, I believe that to show that "The experimental results clearly indicate that our proposed bridged transfer framework and novel generation technique, dubbed 'dataset style inversion,' are responsible for the performance gains.", one should test using pre-trained generative models with similar scale datasets as the baseline (ideally Imagenet-1k, which are available), making an apples-to-apples comparison.
>
> This would allow to disentangle the positive effects of 1) using large-scale data (which is well-known and unsurprising) and 2) sampling from a generative model and training with it (what is studied in the paper). Because of this, it is unclear what gains reported on the experimental section come from 1 or 2, and is misleading to attribute all the gains to 2, the method proposed. This experimental setup is the one tested in previous papers like [1] or Generative models as a data source for multiview representation learning, Jahanian et al.

---

### Official Review · Reviewer_e2MK · 2023-10-29

**Soundness:** 3 good
**Presentation:** 3 good
**Contribution:** 3 good
**Rating:** 6
**Confidence:** 4

**Summary:**

The paper explores the usability of a text-to-image diffusion model (Stable Diffusion) for generating synthetic images to improve transfer learning from ImageNet to different downstream datasets. Different factors of the transfer are compared, for example comparing mixing synthetic images with the downstream data compared to first fine-tuning on synthetic images followed by fine-tuning on the downstream data (bridged transfer). Experiments also show how using mixup in the fine-tuning on synthetic images improves performance, as well as reinitializing the final FC layers in the model when fine-tuning on the downstream data. For the data generation, a variant of textual inversion is explored, where the style of the entire dataset is inverted and can be used to guide the generation of the synthetic dataset.

Experiments on 10 different downstream datasets show that synthetic data adds most value in the bridged transfer with mixup and reinit, and that style inversion is also valuable. Furthermore, different sizes of synthetic dataset is tested, showing distinct improvements with size in the tested regime of dataset sizes, and also for few-shot learning. Experiments are performed using a ResNet-18, but there are also some experiments validating the performance with ResNet-50 and ViTs.

**Strengths:**

+ Promising results for synthetic training data in transfer learning
+ Comprehensive experiments on multiple datasets

More details in questions/comments below.

**Weaknesses:**

- Some more baseline results would be valuable
- There are no comparison of bridged transfer vs. mixing when style inversion is performed

More details in questions/comments below.

**Questions:**

* The main body of experiments (for example when comparing transfer learning strategies), are performed with only text prompts as conditioning for generating synthetic data. As this is expected to generate large domain gaps with the downstream datasets, it is not surprising that the mixing of real and synthetic images performs worst. How would this compare if the style inversion was included, which reduces the domain gap between synthetic and downstream data? Or using some other alignment mechanisms such as textual inversion, dreambooth [1] or diffusion inversion [2].
	[1] Ruiz, N., Li, Y., Jampani, V., Pritch, Y., Rubinstein, M., & Aberman, K. (2023). Dreambooth: Fine tuning text-to-image diffusion models for subject-driven generation. In Proceedings of the IEEE/CVF Conference on Computer Vision and Pattern Recognition (pp. 22500-22510).
	[2] Zhou, Y., Sahak, H., & Ba, J. (2023). Training on Thin Air: Improve Image Classification with Generated Data. arXiv preprint arXiv:2305.15316.

* Comparisons are made to vanilla transfer (without synthetic images) and mixing synthetic and downstream data. However, it would be of interest to also add comparisons to: 1) no transfer, training from scratch on downstream data, to show the benefit of transferring, and 2) pre-training on only synthetic data, to demonstrate if there are gains in doing the 2-stage (bridged transfer), or if it's enough to only pre-train on synthetic data.

* There should also be comparisons to vanilla transfer with FC reinit, which is not included in, e.g., Table 2 and Table 6. It seems that this makes a large difference, but it is only tested for the bridged transfer, not for the vanilla transfer.

* It would be of interest to demonstrate examples of the dataset style inversion, to show how this is better aligned with the downstream dataset. Perhaps also some quantification of the distributional difference compared to real data, e.g. by means of FID or such.

* The circle plots in Fig. 2 and Fig. 8 are not very intuitive to interpret. How are these produced? There is a normalization with vanilla transfer, but how are the axes shifted? For example, in Fig. 2 it seems that axes are also scaled to show mixed transfer on the inner ring, except for the Cars dataset. It would be easier to interpret the results on, e.g., bar plots or similar.

* It is a bit unclear what strategy has been used in figures. For example, in Fig. 2 the performance degrades with bridged transfer on DTD and Dogs, so I assume this is not with the bridged transfer++? However, in Fig. 3 and Fig. 9 bridged transfer performs equal or better on all datasets. So, here the method has been changed to the bridged transfer++? Please clarify around this.

---

> ### Author Response · Authors · 2023-11-21
> **Reply to Reviewer e2MK （Part 1)**
>
> Thank you for your positive feedback on our work. Please check our detailed response below.
>
> Q1: The main body of experiments (for example when comparing transfer learning strategies), are performed with only text prompts as conditioning for generating synthetic data. As this is expected to generate large domain gaps with the downstream datasets, it is not surprising that the mixing of real and synthetic images performs worst. How would this compare if the style inversion was included, which reduces the domain gap between synthetic and downstream data? Or using some other alignment mechanisms such as textual inversion, dreambooth [1] or diffusion inversion [2].
>
> A1: Thank you for your question. We agree with the reviewer that the mixed transfer may benefit from DSI synthetic data. To validate this, we train the mixed transfer using either the DSI synthetic data. The results are shown below for the Aircraft and the Pets dataset.
>
> | Transfer Method    | Syn Method      | Aircraft | Pets |
> |--------------------|-----------------|----------|------|
> | Vanilla Transfer   | N/A             | 79.7     | 91.3 |
> | Mixed Transfer     | Single Template | 70.7     | 82.5 |
> | Mixed Transfer     | DSI             | 81.8     | 91.2 |
> | Bridged Transfer++ | DSI             | 85.8     | 91.9 |
>
> We find that mixed transfer with DSI inversion indeed improves the performance of mixed transfer with Single Template synthesis. However, when compared to Bridged Transfer++, this method still has lower accuracy. In our view, the mixed transfer can be viewed as a type of data augmentation, which needs careful tuning to not differentiate between original images too much. Yet Bridged Transfer is a type of pre-training that has looser requirements in domain gap compared to data augmentations.
>
> Q2: Comparisons are made to vanilla transfer (without synthetic images) and mixing synthetic and downstream data. However, it would be of interest to also add comparisons to: 1) no transfer, training from scratch on downstream data, to show the benefit of transferring, and 2) pre-training on only synthetic data, to demonstrate if there are gains in doing the 2-stage (bridged transfer), or if it's enough to only pre-train on synthetic data.
>
>
> A2: Thank you for your suggestions. We agree that adding these two experiments would benefit the overall understanding of our paper. We perform experiments on Pets and Aircraft datasets, (more datasets will be added in the future). Here are the results showing (1) training from scratch, (2) vanilla transfer from ImageNet, (3) vanilla transfer from synthetic data (not ImageNet pre-trained), (4) Bridged transfer from synthetic data.
>
> It can be found that training scratch has the lowest accuracy, and both ImageNet pretrained or synthetic data pretrained models would benefit the transfer learning. Adding those two factors together leads to our Bridged transfer results, which have the highest performance.
>
> | Method                                | Aircraft | Pets |
> |---------------------------------------|----------|------|
> | Training from scratch                 | 66.3     | 68.5 |
> | Vanilla transfer from ImageNet        | 79.7     | 91.3 |
> | Vanilla transfer from Synthetic data  | 83.0     | 84.5 |
> | Bridged transfer from Synthetic data  | 85.2     | 91.8 |
>
>
> Q3: There should also be comparisons to vanilla transfer with FC reinit, which is not included in, e.g., Table 2 and Table 6. It seems that this makes a large difference, but it is only tested for the bridged transfer, not for the vanilla transfer.
>
> A3: Since the ImageNet dataset has 1000 classes, which is different from the target datasets in transfer learning, we have to reinitialize the classifier in vanilla transfer. In other words, the vanilla transfer is inherently equipped with FC reinit.
>
>
> Q4: It would be of interest to demonstrate examples of the dataset style inversion, to show how this is better aligned with the downstream dataset. Perhaps also some quantification of the distributional difference compared to real data, e.g. by means of FID or such.
>
> A4: Thanks for your comment. We have added a figure in Appendix D in the revised version. Meanwhile, we have measured the FID score and demonstrated them in the table below.
>
> | Synthesis Method | Aircraft FID | DTD FID |
> |------------------|--------------|---------|
> | Single Template  | 19.5         | 28.3    |
> | DSI              | 10.1         | 14.3    |

---

> ### Author Response · Authors · 2023-11-21
> **Reply to Reviewer e2MK （Part 2)**
>
> Q5: The circle plots in Fig. 2 and Fig. 8 are not very intuitive to interpret. How are these produced? There is a normalization with vanilla transfer, but how are the axes shifted? For example, in Fig. 2 it seems that axes are also scaled to show mixed transfer on the inner ring, except for the Cars dataset. It would be easier to interpret the results on, e.g., bar plots or similar.
>
> A5: Thanks for your comment. We apologize for not mentioning the axis normalization.  For Figure 2, we normalize each axis so that the vanilla transfer is fixed to the 3rd ring. Under this normalization, we scale the axis so that neither the mixed transfer goes inside the 1st ring nor the bridged transfer goes outside the 4th ring. We have changed Fig. 2 to prevent confusion on mixed transfer, where we normalize the mixed transfer to the 1st ring.
>
> For Figure 8, we normalize each axis so that the vanilla transfer is fixed to the 2nc ring. Then, we scale the axis so that the bridged transfer is fixed to the 4th ring unless the difference is less than 2%. If the difference is less than 2%, we will scale the interval to a 1% accuracy difference. We understand that It would be easier to interpret the results with bar plots, however, it may significantly increase the space of the plots. Therefore, we refrain from doing so.
>
>
> Q6: It is a bit unclear what strategy has been used in figures. For example, in Fig. 2 the performance degrades with bridged transfer on DTD and Dogs, so I assume this is not with the bridged transfer++? However, in Fig. 3 and Fig. 9 bridged transfer performs equal or better on all datasets. So, here the method has been changed to the bridged transfer++? Please clarify this.
>
> A6: Thanks for your comment. We’d like to clarify that Fig 3 and Fig 9 actually report the training accuracy rather than the test accuracy. We did not employ Bridged Transfer++ in those two figures. We intended to show that Bridge Transfer has better convergence during training, but suffers from lower test accuracy due to overfitting.

---

> > ### Comment · Reviewer_e2MK · 2023-11-22
> >
> > Thanks for the detailed responses to the questions and comments. These clarify really well how the proposed transferring strategy could be valuable option for combining the benefits of pre-training on ImageNet and synthetic data, at least on the Aircraft and Pets datasets. Also nice to see examples of the style inversion, in image examples and comparisons in terms of FID. I also think it is important to clarify around the details of the plots in the main paper, since there is some potential sources of confusion around these. Apart from this, I believe the paper contributes with a valuable exploration into the benefits of using synthetic data to promote transfer learning.

---

### Official Review · Reviewer_HV3t · 2023-10-29

**Soundness:** 4 excellent
**Presentation:** 4 excellent
**Contribution:** 4 excellent
**Rating:** 8
**Confidence:** 5

**Summary:**

Synthetic image data is quite important for model training/fine-tuning/transfer learning under a series of practical problems such as data shortage, privacy, IP considerations, etc. This work delves into the generation and utilization of synthetic images derived from text-to-image generative models in facilitating transfer learning paradigms. This work observed that, despite the high visual fidelity of the generated images, their naive incorporation into existing real-image datasets does not consistently enhance model performance due to the inherent distribution gap between synthetic and real images. To address this issue, this paper introduced a novel two-stage framework called bridged transfer, which initially employs synthetic images for fine-tuning a pre-trained model to improve its transferability and subsequently uses real data for rapid adaptation. Alongside, they proposed a dataset style inversion strategy to improve the stylistic alignment between synthetic and real images. Empirical evaluation across 10 different datasets and 5 distinct models demonstrates their consistent improvements.

**Strengths:**

- Novel observation: this paper firstly observed that the naive incorporation of synthesized images into existing real-image datasets does not consistently enhance model performance due to the inherent distribution gap between synthetic and real images by examining three key facets—data utilization, data volume, and data generation control—across multiple downstream datasets. This observation and all the listed key takeaways are very inspiring for the community.
- Effective framework: To address this issue, this paper then introduced a novel two-stage framework called bridged transfer, which initially employs synthetic images for fine-tuning a pre-trained model to improve its transferability and subsequently uses real data for rapid adaptation.
- Effective strategy to further enhance performance: this paper further proposed dataset style inversion strategy to improve the stylistic alignment between synthetic and real images.
- Extensive and solid experiments: this paper evaluated across 10 different datasets and 5 distinct models, demonstrating consistent improvements.

**Weaknesses:**

- The high-quality synthetic images can always bring multi-fold benefits (cost reduction on real data collection, data shortage, privacy, IP, etc), it would be beneficial for understanding this work’s importance by clearly explaining or numerically analyze how this work can relate to these benefits.
- More discussion is expected to better understand the scope and possibility of observed phenomenon, proposed framework, and strategy. For example, beyond the transfer learning paradigms, will this observation also hold in other learning paradigms? Beyond text-to-image generative models, will the idea of bridged transfer also work?

**Questions:**

1. Beyond the transfer learning paradigms, will this observation also hold in other learning paradigms?
2. Beyond text-to-image generative models, will the idea of bridged transfer also work?

---

> ### Author Response · Authors · 2023-11-21
> **Reply to Reviewer HV3t**
>
> Thank you for your positive feedback on our work. Please check our detailed response below.
>
> Q1: More analysis on benefits of synthetic images
>
> A1: We quantitatively convert the effect of synthetic data to the effect of real data by studying this question in few-shot learning scenarios. We run the vanilla transfer from 1-shot to 40-shot one by one and match the accuracy of the bridged transfer. Therefore, we can evaluate how much data the vanilla transfer would need to realize the same accuracy as the bridged transfer. The difference between vanilla and bridged transfer is the gains of synthetic data. The results for the Aircraft dataset of ResNet-18 are shown below:
>
> | Accuracy | Bridged Transfer (w. 3k images / cls) | Vanilla Transfer |
> |----------|---------------------------------------|------------------|
> | 25.1     | 1-shot                                | 6-shot           |
> | 45.0     | 4-shot                                | 14-shot          |
> | 58.8     | 8-shot                                | 24-shot          |
> | 71.4     | 16-shot                               | 40-shot          |
>
>
> Significantly, our findings reveal that augmenting a dataset with 3,000 synthetic images per class can be as effective as acquiring 24 new real images per class, an impactful insight considering our initial dataset comprises just 16 images per class. From a cost perspective, this is noteworthy: commercial image procurement may range from 1 to 3 dollars per image, whereas leveraging stable diffusion to generate 3,000 images entails a mere $1.50, as documented by AWS's cost analysis (https://aws.amazon.com/blogs/machine-learning/maximize-stable-diffusion-performance-and-lower-inference-costs-with-aws-inferentia2/). This cost efficiency could represent significant savings for researchers and commercial entities alike.
>
> From the standpoint of privacy protection, synthetic image generation offers a substantial advantage. Traditional methods involving web-scraped or manually collected images mandate rigorous examination for personal identifiers and sensitive content, a process that can be both time-consuming and prone to error. In contrast, synthetic data not only circumvents many of these privacy concerns but also enables a mathematical assessment of potential risks. Studies such as [1, 2] have explored the concept of Differentially Private Stable Diffusion (DP-SD), suggesting that models trained on DP-SD-generated data inherit the framework's privacy-preserving attributes, thereby enhancing the overall security of the information.
>
> [1] Lin, Z., Gopi, S., Kulkarni, J., Nori, H. and Yekhanin, S., 2023. Differentially Private Synthetic Data via Foundation Model APIs 1: Images. arXiv preprint arXiv:2305.15560.
>
> [2] Sehwag, V., Panda, A., Pokle, A., Tang, X., Mahloujifar, S., Chiang, M., Kolter, J.Z. and Mittal, P., 2023. Differentially Private Generation of High Fidelity Samples From Diffusion Models.
>
> Q2: Beyond the transfer learning paradigms, will this observation also hold in other learning paradigms?
>
> A2: Yes, we can apply synthetic images to a randomly initialized model (no real data pre-trained), and then transfer to target datasets. Here, we test the performance on Aircraft and Pets datasets. The results are shown below:
>
> | Method                                | Aircraft | Pets |
> |---------------------------------------|----------|------|
> | Training from scratch                 | 66.3     | 68.5 |
> | Vanilla transfer from ImageNet        | 79.7     | 91.3 |
> | Vanilla transfer from Synthetic data  | 83.0     | 84.5 |
> | Bridged transfer from Synthetic data  | 85.2     | 91.8 |
>
> It can be found that our method can effectively improve the performance of a randomly initialized model. On the Aircraft dataset, the synthetic data even outperforms the ImageNet pretrained models.
>
> Q3: Beyond text-to-image generative models, will the idea of bridged transfer also work?
>
> A3: Thanks for the question. Generally, we think bridged transfer will also work on other types of synthetic data, like GAN-based synthetic data. However, given the time limit in the rebuttal period and the synthesis model is a little out of our scope, we'd like to leave it as our future work.

---

> > ### Comment · Reviewer_HV3t · 2023-11-22
> > **Official Comments by Reviewers**
> >
> > Using synthetic images for model training/fine-tuning/transfer learning is quite an interesting and important topic. The reviewer would like to thank the detailed responses from the authors. Most of my concerns have been addressed during the rebuttal, therefore I will keep my original score and vote for the acceptance.

---

### Official Review · Reviewer_RCjJ · 2023-11-02

**Soundness:** 2 fair
**Presentation:** 3 good
**Contribution:** 2 fair
**Rating:** 6
**Confidence:** 4

**Summary:**

The paper proposes an approach "Bridged Transfer" to improve transfer learning in computer vision by augmenting the dataset using text-to-image generative models. The paper introduces a two stage approach for transfer learning - 1) creating a synthetic dataset for target task using text-to-image generative models 2) training the backbone network using this synthetic dataset followed by reinitializing the classifier weights in order to finally train the classifier via target real dataset. The paper shows using naive mixing of real and synthetic data fails to improve transfer learning. The paper also proposes to reduce the distribution gap between the generated dataset and real target dataset using dataset style inversion technique that stylizes and aligns the generated dataset with the real target dataset. The paper empirically shows improvement of this transfer learning algorithm for multiple target classification tasks.

**Strengths:**

The paper explores a transfer learning framework that augments the target datasets with generated datasets derived from text-to-image models. The proposed approach “Bridged Transfer” achieves improvement on various image classification tasks (~ 1%- 6%) along with few-shot classification tasks. The paper provides detailed ablation experiments to analyze the various components of Bridged transfer algorithm. Ablation experiments regarding the size of synthetic dataset sheds light on the benefit of utilizing this synthetic dataset (for eg higher volume of synthetic datasets leads to better performance). The paper is well-written and easy to follow. The paper provides code for reproducibility.

**Weaknesses:**

It would be helpful for the reader to get a better understanding of the following:

1. It would interesting to have the approach proposed in paper “IS SYNTHETIC DATA FROM GENERATIVE MODELS READY FOR IMAGE RECOGNITION?” as a baseline (let’s say this baseline as Transfer1). Specifically, it would be interesting to see the following:
    1. Performance of Transfer1 on the same setting (i.e. having same generative text2image model, same evaluation datasets and using the proposed approaches of dataset filtering RF, RG to close the domain gap)
    2. Performance of Bridged transfer on pre-trained CLIP models and then comparison with Transfer1 on the same proposed evaluation setting introduced by Transfer1.
    3. The idea of the above 2 suggestions is to glean out what additional information we can learn on top of the findings discovered by Transfer1 approach.
    4. Also, it might be helpful for the reader, if there was a subsection/paragraph that describes the difference between Bridged transfer and Transfer1. For example Bridged transfer uses Dataset style inversion to reduce the domain gap whereas Transfer1 leans on filtering approaches to create a higher quality dataset.
2.  It might be helpful to see some qualitative results on the benefits of data style inversion (DSI). It can visually show that the domain gap of synthetic datasets with target dataset decreases (for e.g. comparison between with and without DSI module for the same text prompt in the context of some target datasets)
3. It might improve the readability of the paper if Bridged Transfer+FC Reinit & mixup is also included in the figure 2.

**Questions:**

It would be helpful if the paper can answer the following questions:
1. How much training time/wall-clock time and memory resources does it take for dataset style inversion (DSI) for each dataset to optimize the S* token? For e.g. it’s mentioned that it takes 20k training iterations for Aircraft dataset.
2. Are there any qualitative reasons that for datasets like DTD and Pets, the improvement is kind of limited ? Does the synthetic dataset has higher domain gap or is this a fine grained task for which text-to-image models can’t generate enough informative samples. Also, it would be interesting to see the impact of synthetic data volume increase for these classes similar to Fig 5(a).
3. Is Bridged Transfer++ equal to “Bridged Transfer + FC Reinit & mixup” or “Bridged Transfer + FC Reinit & mixup + DSI” ? In table 3, does Single Template correspond to “Bridged Transfer + FC Reinit & mixup” ? If yes, why are the number for DTD not matching up between Table 2  (72.3 +- 0.3) and Table 3 (71.5+-0.4) while the numbers does match for rest of the dataset like Aircraft, Cars, Foods and SUN397 ?
4. In table 3, what does Single Template correspond to (which Bridged Transfer version and what’s the network architecture)?

---

> ### Author Response · Authors · 2023-11-21
> **Reply to Reviewer RCjJ (Part 1)**
>
> Thank you for your comments and suggestions to improve our paper. We would like to reply to your questions and suggestions in detail.
>
> Q1: Performance of Transfer1 on the same setting (i.e. having same generative text2image model, same evaluation datasets and using the proposed approaches of dataset filtering RF, RG to close the domain gap)
>
> Q2: Performance of Bridged transfer on pre-trained CLIP models and then comparison with Transfer1 on the same proposed evaluation setting introduced by Transfer1.
>
> Q3: The idea of the above 2 suggestions is to glean out what additional information we can learn on top of the findings discovered by Transfer1 approach.
>
>
>
> A1/2/3: We are grateful for the recommendation to juxtapose our method with Transfer 1. While acknowledging the importance of Transfer 1 in utilizing synthetic data for image recognition, we aim to delineate the distinctions and similarities between our approaches.
> We must emphasize that Transfer 1 utilizes classifier tuning tailored for the CLIP model; conversely, our work concentrates on whole network tuning applicable to a spectrum of models, such as ResNets. It is crucial to recognize the significant differences in pre-training, fine-tuning, and application methods between the models assessed by Transfer 1 and those in our study. For instance, it has been observed that fine-tuning an entire CLIP model may detrimentally impact its robustness and accuracy [1]. Consequently, not all insights from Transfer 1 are directly transferrable to our proposed bridge transfer frameworks, and vice versa—a point we will substantiate through our experiments. A direct comparison between the tuning of CLIP and general models falls beyond the scope of Transfer 1 and our work.
>
> In this study, we delve into transfer learning for general models, as this scenario is not only prevalently employed in practical settings but also merits investigation to comprehend how synthetic data can enhance these applications.
>
> [1] Wortsman, M., Ilharco, G., Kim, J. W., Li, M., Kornblith, S., Roelofs, R., ... & Schmidt, L. (2022). Robust fine-tuning of zero-shot models. In Proceedings of the IEEE/CVF Conference on Computer Vision and Pattern Recognition (pp. 7959-7971).

---

> ### Author Response · Authors · 2023-11-21
> **Reply to Reviewer RCjJ (Part 2)**
>
> While it is challenging to draw an apple-to-apple comparison between Transfer 1 and our method, we have endeavored to adapt both for comparison: 1) We employed Stable Diffusion with RF and RG for Transfer 1, based on its official code, and 2) We performed bridged transfer (limited to the classifier) with CLIP. It's important to clarify that this experiment is not about directly contrasting the performance of Transfer 1 and our method in their unaltered states, but rather to understand how they operate within different configurations. Due to the time limit in the rebuttal period, we select Aircraft and Pets datasets to test the performance.
>
> | Model     | Synthesis      | Transfer Method    | Aircraft 1-shot | Aircraft 8-shot | Pets 1-shot  | Pets 8-shot |
> |-----------|----------------|--------------------|-----------------|-----------------|--------------|-------------|
> | CLIP      | N/A            | Vanilla Transfer   | 20.2            | 32.7            | 86.2         | 87.9        |
> | CLIP      | Basic          | Mixed (Transfer 1) | 21.9            | 31.8            | 87.1         | 87.8        |
> | CLIP      | Basic          | Bridged Transfer   | 20.9            | 31.7            | 87.5         | 87.8        |
> | CLIP      | Real Guidance  | Mixed (Transfer 1) | 22.6            | 33.4            | 87.5         | 89.5        |
> | CLIP      | Real Guidance  | Bridged Transfer   | 21.8            | 32.8            | 87.3         | 89.3        |
> | ResNet-18 | N/A            | Vanilla Transfer   | 6.5             | 32.2            | 38.5         | 80.4        |
> | ResNet-18 | Basic          | Bridged Transfer++ | 25.0            | 58.8            | 54.8         | 87.5        |
>
> Based on the results, we have several common observations with Transfer 1 and several new findings on top of it.
>
> 1. In few-shot recognition, Real Guidance synthesis is always better than Basic synthesis, which is also observed in Transfer 1.
>
> 2. However, in 8-shot setting, we found that Basic synthesis + Mixed (Transfer 1) cannot improve the vanilla transfer, this is similar to our paper’s finding that regular synthetic data has a distribution shift and may lead to accuracy decreases in mixed transfer.
>
> 3. The bridged transfer does not always perform better than the mixed transfer using the Transfer 1 framework (CLIP, classifier fine-tuning) which is also discovered by Transfer 1 in Table 7.
> We think the reasons are twofold: (1) Transfer 1 uses classifier tuning of the CLIP model, while our pipeline uses whole network tuning. In our paper, we found that synthetic data may lead to bias in the final classifier layer, therefore, we employ the FC reinitialization in bridged transfer++. In Transfer 1, since only the classifier is allowed to be tuned, it is not surprising that bridged transfer may have lower performance than mixed transfer when only optimizing the classifier. (2) Transfer 1 uses different batch sizes to balance the synthetic data and real data (512 for synthetic data, 32 for real data) so that the network weighs more in real data to prevent distribution shifts. This is why Transfer 1 needs to freeze BN, while we do not need to freeze BN layers in bridged transfer.
> Nonetheless, when we apply our bridged transfer approach to ResNet-18, it yields enhanced accuracy on the challenging fine-grained classification dataset 'Aircraft', despite ResNet-18 being significantly smaller than CLIP. Moreover, Transfer 1 exhibits superior performance on the 'Pets' dataset. These outcomes corroborate our earlier assertion: although Transfer 1 and our method focus on different models and scenarios, they both effectively demonstrate the potential of synthetic data to boost accuracy for their respective settings and tasks.
>
> We hope our response can sufficiently address your concerns.

---

> ### Author Response · Authors · 2023-11-22
> **Reply to Reviewer RCjJ (Part 3)**
>
> Q4: Also, it might be helpful for the reader, if there was a subsection/paragraph that describes the difference between Bridged transfer and Transfer1. For example, Bridged transfer uses Dataset style inversion to reduce the domain gap whereas Transfer1 leans on filtering approaches to create a higher-quality dataset.
>
> A4: Thanks for your suggestion, we have added a section in Appendix E showing the difference between our work and Transfer 1. In the previous experiments, we conducted experiments under the Transfer 1 framework and we believe these two papers share similar observations in leveraging the synthetic data. In addition to Transfer 1, our work studied how to more effectively utilize synthetic in the general transfer learning framework. We find that synthetic data promotes feature representation learning but may bias the FC layer. We will add these inspiring discussions to the main paper when we have more dataset results.
>
> Q5: It might be helpful to see some qualitative results on the benefits of data style inversion (DSI). It can visually show that the domain gap of synthetic datasets with target dataset decreases (for e.g. comparison between with and without DSI module for the same text prompt in the context of some target datasets)
>
> A5: Thanks for your comment, we have added a figure in Appendix D to show the effect of DSI. Please check it.
>
> Q6: It might improve the readability of the paper if Bridged Transfer+FC Reinit & mixup is also included in figure 2.
>
> A6: Thank you for your suggestion. We have added the Birdged Transfer++ (Bridged Transfer+FC Reinit & mixup) results into Fig 2.
>
> Q7 How much training time/wall-clock time and memory resources does it take for dataset style inversion (DSI) for each dataset to optimize the S* token? For e.g. it’s mentioned that it takes 20k training iterations for the Aircraft dataset.
>
> A7: For all datasets, we optimize the token for 20k training iterations (3 hours on one A100 GPU). Hence, the training wall-clock time is the same across different datasets. Note that when compared to class-wise textual inversion, the speed-up ratio will be correlated with a number of classes in that dataset.
>
> Q8: Are there any qualitative reasons that for datasets like DTD and Pets, the improvement is kind of limited ? Does the synthetic dataset has higher domain gap or is this a fine grained task for which text-to-image models can’t generate enough informative samples. Also, it would be interesting to see the impact of synthetic data volume increase for these classes similar to Fig 5(a).
>
> A8: During our experiments, we found that there could be two reasons that make DTD and Pets hard to improve with synthetic data. (1) The domain gap between synthetic and real data is too huge, or the label cannot describe the class very well. For example, in the DTD dataset, the label name is the adjective of each texture, e.g., “banded, braided”, which can correspond to lots of different banded textures and braided textures. In this case, we have to use DSI to mitigate the gap. (2) The second reason is the original datasets contain enough data samples for recognition, adding much more synthetic data might lead to severe overfitting, e.g. Pets. In this case, we can observe much more improvements in few-shot transfer, while moderate improvements in full-shot transfer.
>
> Q9:Is Bridged Transfer++ equal to “Bridged Transfer + FC Reinit & mixup” or “Bridged Transfer + FC Reinit & mixup + DSI” ? In table 3, does Single Template correspond to “Bridged Transfer + FC Reinit & mixup” ? If yes, why are the number for DTD not matching up between Table 2 (72.3 +- 0.3) and Table 3 (71.5+-0.4) while the numbers does match for rest of the dataset like Aircraft, Cars, Foods and SUN397 ?
>
> Q10: In table 3, what does Single Template correspond to (which Bridged Transfer version and what’s the network architecture)?
>
> A9/10: Thank you for your question. We apologize for the unclear description and wrong numbers in Table 3. In Table 3, the Single Template uses ResNet-18 as the architecture and both Single Template and DSI use Bridged Transfer ++, which is Bridged Transfer with FC Reinit and mixup. We revisited our experiments and found that we forgot to add a mixup loss function when running DTD datasets. We have changed the accuracy to new ones.

---

> > ### Comment · Reviewer_RCjJ · 2023-11-22
> > **Thank you for the response**
> >
> > Thank you for providing explanation for my questions. I believe this paper could contribute in opening a discussion about how we can use synthetic datas from foundation models for downstream tasks in the research community. Accordingly, I have increased my score to 6.

---

### Meta-Review · Program_Chairs · 2023-12-11

**Metareview:**

The paper explores a direction of generating synthetic images using text-to-image diffusion models to improve transfer learning from ImageNet to different downstream datasets, and name this approach "bridged transfer".  Some reviewers appreciated the interesting topic, the proposed approach, and the comprehensive experiments. However, concerns were raised about the experiments, the clarity of certain figures, and the potential bias towards foundation models.

PC/SAC comment: after calibration and downweighting inflated and non-informative reviews, the decision is to reject at this time.

**Justification For Why Not Higher Score:**

Please see meta-review

**Justification For Why Not Lower Score:**

Please see meta-review

---

### Decision · Program_Chairs · 2024-01-16

Reject